# The uncharacterized protein FAM47E interacts with PRMT5 and regulates its functions

Baskar Chakrapani[1,*], Mohd Imran K Khan[1,*], Rajashekar Varma Kadumuri[2], Somlee Gupta[1], Mamta Verma[1], Sharad Awasthi[1], Gayathri Govindaraju[3], Arun Mahesh[1], Arumugam Rajavelu[3], Sreenivas Chavali[2], Arunkumar Dhayalan[1]

Protein arginine methyltransferase 5 (PRMT5) symmetrically dimethylates arginine residues in various proteins affecting diverse cellular processes such as transcriptional regulation, splicing, DNA repair, differentiation, and cell cycle. Elevated levels of PRMT5 are observed in several types of cancers and are associated with poor clinical outcomes, making PRMT5 an important diagnostic marker and/or therapeutic target for cancers. Here, using yeast two-hybrid screening, followed by immunoprecipitation and pull-down assays, we identify a previously uncharacterized protein, FAM47E, as an interaction partner of PRMT5. We report that FAM47E regulates steady-state levels of PRMT5 by affecting its stability through inhibition of its proteasomal degradation. Importantly, FAM47E enhances the chromatin association and histone methylation activity of PRMT5. The PRMT5–FAM47E interaction affects the regulation of PRMT5 target genes expression and colony-forming capacity of the cells. Taken together, we identify FAM47E as a protein regulator of PRMT5, which promotes the functions of this versatile enzyme. These findings imply that disruption of PRMT5–FAM47E interaction by small molecules might be an alternative strategy to attenuate the oncogenic function(s) of PRMT5.

## Introduction

Arginine methylation is a widely prevalent, important posttranslational modification affecting various cellular processes (Peng & Wong, 2017). Protein arginine methyltransferase 5 (PRMT5) belongs to type II methyltransferases that symmetrically dimethylate the arginine residues of the target proteins (Bedford & Clarke, 2009). PRMT5 plays an important role in the regulation of gene expression, splicing, chromatin remodeling, cell differentiation, and development (Stopa et al, 2015). PRMT5 participates in epigenetic regulation of chromatin structure and gene expression by introducing symmetric dimethylation at arginine 3 of histone 4 (H4R3me2s), arginine 2 and 8 of histone 3 (H3R2me2s and H3R8me2s) and arginine 3 of histone 2A (H2AR3me2s) (Pollack et al, 1999; Branscombe et al, 2001; Pal et al, 2004; Ancelin et al, 2006; Migliori et al, 2012). Apart from histones, PRMT5 methylates and regulates the function of a wide variety of non-histone proteins involved in diverse biological processes such as (i) DNA repair: FEN1 (Guo et al, 2010); (ii) transcription: p53 (Jansson et al, 2008; Scoumanne et al, 2009), SPT5 (Kwak et al, 2003), E2F1 (Cho et al, 2012), MBD2 (Tan & Nakielny, 2006), HOXA9 (Bandyopadhyay et al, 2012), NF-κB (Harris et al, 2016), SREBP1 (Liu et al, 2016), FOXP3 (Nagai et al, 2019), BCL6 (Lu et al, 2018), Tip60 (Clarke et al, 2017), and RNAPII (Zhao et al, 2016); (iii) splicing: Sm proteins (Friesen et al, 2001; Meister et al, 2001), (iv) translation: ribosomal protein S10 (Ren et al, 2010) and hnRNP A1 (Gao et al, 2017), (v) signaling: EGFR (Hsu et al, 2011), PDGFRα (Calabretta et al, 2018), and CRAF (Andreu-Perez et al, 2011); (vi) organelle biogenesis: GM130 (Zhou et al, 2010); and (vii) stress response: G3BP1 (Tsai et al, 2016) and LSM4 (Arribas-Layton et al, 2016).

PRMT5 plays a critical role in the differentiation of primordial germ cells, nerve cells, myocytes, and keratinocytes (Ancelin et al, 2006; Dacwag et al, 2007, 2009; Huang et al, 2011; Chittka et al, 2012; Kanade & Eckert, 2012; Paul et al, 2012). Notably, the knockout of PRMT5 leads to embryonic lethality, reflecting its essentiality for development and survival (Tee et al, 2010). From a pathological stand point, aberrant expression of human PRMT5 is observed in diverse cancer types (Stopa et al, 2015; Xiao et al, 2019). Elevated expression of PRMT5 in epithelial ovarian cancer and non-small cell lung cancer is associated with poor clinical outcomes and patient survival (Bao et al, 2013; Győrffy et al, 2013; Stopa et al, 2015). Depletion of PRMT5 inhibits cell proliferation, clonogenic capacity of the cells, and improves the prognosis of cancer patients making PRMT5 an important target for cancer therapy (Pal et al, 2004; Scoumanne et al, 2009; Wei et al, 2012; Chung et al, 2013; Morettin et al, 2015; Yang et al, 2016; Banasavadi-Siddegowda et al, 2018; Saloura et al, 2018; Xiao et al, 2019).

[1]Department of Biotechnology, Pondicherry University, Puducherry, India  [2]Department of Biology, Indian Institute of Science Education and Research (IISER) Tirupati, Tirupati, India  [3]Interdisciplinary Biology, Rajiv Gandhi Centre for Biotechnology, Trivandrum, India

Correspondence: schavali@iisertirupati.ac.in; arun.dbt@pondiuni.edu.in
*Baskar Chakrapani and Mohd Imran K Khan contributed equally to this work

The enzymatic activity, substrate specificity, subcellular localization, and functions of PRMT5 is often regulated by its interaction partners (Stopa et al, 2015). For instance, PRMT5 forms a hetero-octameric complex with WD40 repeat protein, MEP50, and the PRMT5–MEP50 complex has higher enzymatic activity than PRMT5 in the unbound state (Friesen et al, 2002; Antonysamy et al, 2012; Ho et al, 2013). PRMT5 interacts with pICln or RioK1 in a mutually exclusive manner and promotes the methylation of Sm proteins or nucleolin, respectively. This highlights that the interaction partners determine the substrate specificity of PRMT5 (Friesen et al, 2001; Meister et al, 2001; Guderian et al, 2011). Interaction of PRMT5 with Menin or COPR5 promotes the recruitment of PRMT5 to the specific promoter regions of chromatin (Lacroix et al, 2008; Paul et al, 2012; Gurung et al, 2013). Blimp1 interacts with PRMT5 and specifies its sub-cellular localization in primordial germ cells (Ancelin et al, 2006). Binding of PRMT5 to interactors such as AJUBA (Hou et al, 2008), JAK kinase (Pollack et al, 1999; Liu et al, 2011), CRTC2 (Tsai et al, 2013), SHARPIN (Tamiya et al, 2018), carbonic anhydrase 6B (Xu et al, 2017), LYAR (Ju et al, 2014), STRAP (Jansson et al, 2008), PHF1 (Liu et al, 2018), CITED2 (Shin et al, 2018), ERG (Mounir et al, 2016), HSP90 (Maloney et al, 2007), CHIP (Zhang et al, 2016), ZNF224 (Cesaro et al, 2009), and AKT (Zhang et al, 2019) engage PRMT5 in diverse cellular processes.

Given the versatile functions of PRMT5 in the cell and its multiple interaction partners and substrates, the identification and characterization of new interaction partners is very important to obtain a comprehensive understanding of the diverse roles of PRMT5 in the cell. To address this, we performed yeast two-hybrid (Y2H) screening to identify new interaction partners of PRMT5 and identified a novel interaction partner FAM47E (family with sequence similarity 47, member E), an hitherto uncharacterized protein. In addition to identifying the new interaction partner of PRMT5; here, we have characterized the functions of FAM47E. We report that FAM47E regulates the stability, chromatin association and methyltransferase activity of PRMT5, with potential implications in normal physiology and in diseases.

## Results

### PRMT5 interacts with FAM47E

To identify new interaction partners of PRMT5, we performed Y2H screening of PRMT5 using universal normalized human cDNA library. Three positive clones were obtained in the initial screening with low stringency selection medium which scores for the expression of two reporter genes. Of these three clones, one of them failed in high stringency selection medium which assesses the expression of four reporter genes and hence it was not considered for further study. Sequencing of the other two positive clones revealed that they code for full length COP9 signaling complex subunit 5 (COPS5 protein) and the C-terminal region of FAM47E isoform 2 (43 amino acid to 295 amino acid). The COPS5 protein is known to interact with GAL4 DNA-binding domain directly and produce false positive results in GAL4-based Y2H screenings (Nordgård et al, 2001; Mohr & Koegl, 2012) and hence we disregarded it for further consideration. The FAM47E gene has three alternative

splice variants, with the isoform 1, the longest being designated as the canonical form in the Uniprot database. However, we obtained FAM47E isoform 2 as putative interaction partner of PRMT5 in our Y2H screening. Hence, we profiled the expression of all the three FAM47E isoforms in HEK293 cells by quantitative RT (qRT)-PCR using isoform specific primers. We found that FAM47E isoform 2 mRNA levels are ~73-folds higher than that of FAM47E isoform 1 and it is ~1,218-folds higher than that of FAM47E isoform 3 suggesting that FAM47E isoform 2 is the predominant isoform in HEK293 cells (Fig S1). Next, we validated PRMT5–FAM47E interaction in Y2H assay by using full-length FAM47E isoform 2 (hereafter referred as FAM47E) with appropriate vector controls. We observed the expression of reporter genes in both low stringency media and high-stringency media only if the Y2H constructs of PRMT5 and FAM47E were co-transformed. We could not detect expression of reporter genes when either of the constructs is co-transformed with the corresponding vector control suggesting that FAM47E is a potential interaction partner of PRMT5 (Fig 1A).

To confirm the PRMT5-FAM47E interaction in vivo, we performed the co-immunoprecipitation (Co-IP) experiments by co-expressing Myc-tagged PRMT5 with GFP or GFP-tagged FAM47E in HEK293 cells. Similarly, we co-expressed HA-tagged FAM47E with GFP or GFP-tagged PRMT5 for reverse Co-IP experiments. We found that GFP-tagged FAM47E efficiently co-precipitated the Myc-tagged PRMT5 in forward Co-IP (Fig 1B) and GFP-tagged PRMT5 co-precipitated the HA-tagged FAM47E in reverse Co-IP (Fig 1C). These observations suggest that PRMT5 could interact with FAM47E in vivo.

We next assessed whether the ectopically expressed GFP-tagged FAM47E could interact with endogenous PRMT5. For this, we overexpressed GFP or GFP-tagged FAM47E in HEK293 cells and performed the Co-IP experiments. We found that GFP-FAM47E efficiently co-precipitated the endogenous PRMT5 suggesting that FAM47E could interact with endogenous levels of PRMT5 (Fig 1D). To assess whether PRMT5 directly interacts with FAM47E, we carried out a GST pull-down assay using recombinant GST-tagged FAM47E and His-tagged PRMT5 proteins. Whereas the control GST protein did not precipitate the His-PRMT5, GST-FAM47E efficiently precipitated His-PRMT5 indicating that PRMT5 interacts with FAM47E directly (Fig 1E). We then performed immunoprecipitation to investigate PRMT5–FAM47E interaction at their endogenous levels in HEK293 cells. Immunoprecipitation experiments were carried out in HEK293 cell lysate using FAM47E antibody or PRMT5 antibody along with the control IgG. We found that immunoprecipitation of FAM47E antibody efficiently co-precipitated the PRMT5 (Fig 1F) and similarly immunoprecipitation of PRMT5 antibody efficiently co-precipitated the FAM47E (Fig 1G). These findings suggest that PRMT5 interacts with FAM47E at their endogenous levels.

Because the WD40 repeat protein, MEP50, interacts and forms a stable hetero-octameric complex with PRMT5 (Friesen et al, 2002; Antonysamy et al, 2012; Ho et al, 2013), we investigated whether FAM47E also interacts with MEP50 in addition to PRMT5. For this, we performed co-immunoprecipitation by co-expressing GFP or GFP-FAM47E with Myc-tagged MEP50 in HEK293 cells and found that FAM47E interacts with MEP50 (Fig S2A). This prompted us to investigate if FAM47E affects the binding of MEP50 with PRMT5. For this, we performed co-immunoprecipitation by co-expressing GFP or GFP-PRMT5 and Myc-tagged MEP50 with or without HA tagged

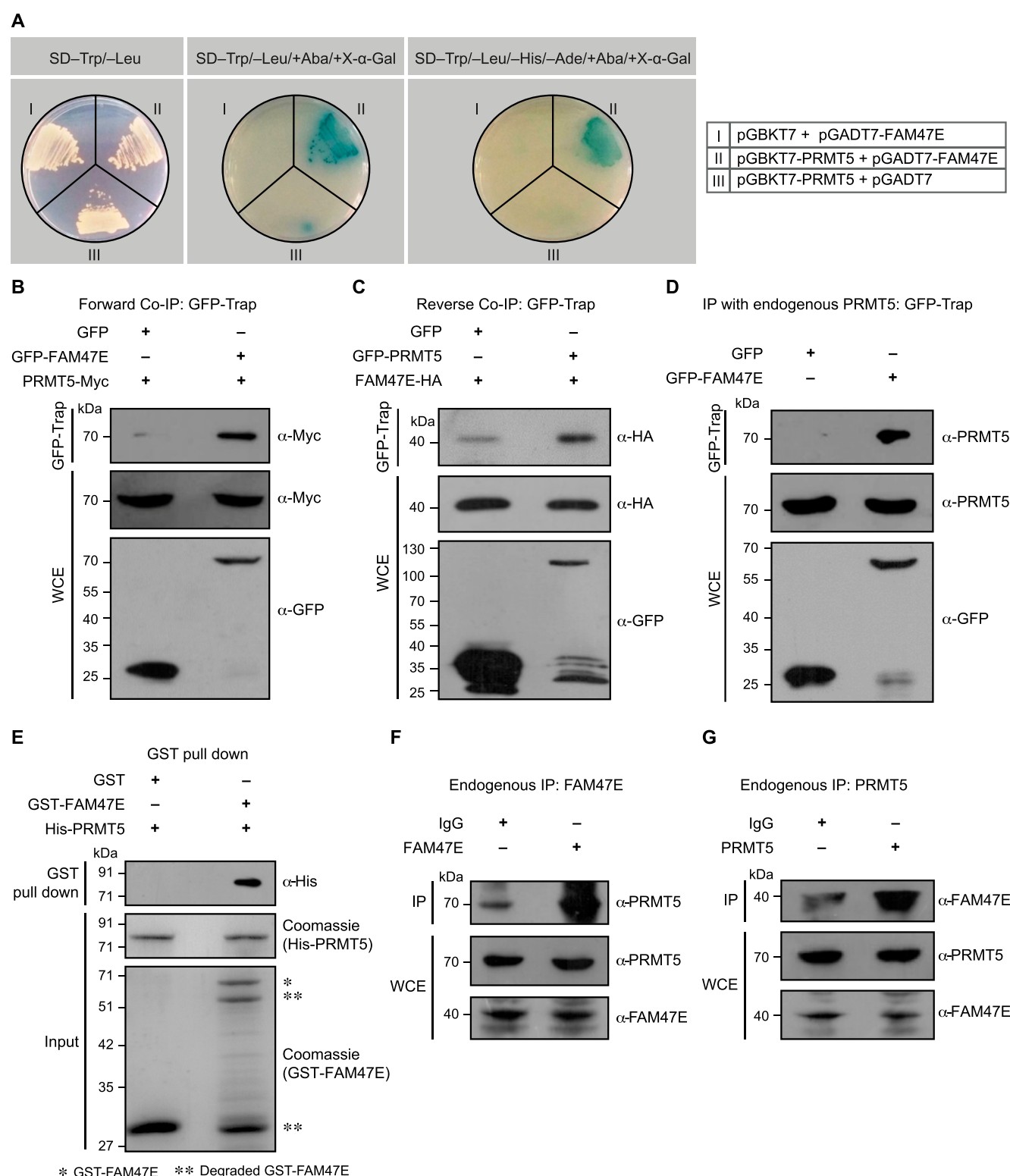

**Figure 1. PRMT5 interacts with FAM47E.**
**(A)** Y2H assay was performed to study the interaction between PRMT5 with FAM47E. The pGBKT7-PRMT5 and PGADT7-FAM47E constructs were allowed to interact with each other or corresponding vector controls. The positive interaction was assessed by the investigating the expression of reporter genes. SD–Trp/–Leu denotes the synthetically defined medium which lacks tryptophan and leucine, SD–Trp/–Leu/+Aba/+X-α-Gal denotes synthetically defined medium which lacks tryptophan and leucine but contains Aureobasidin A and X-α-Gal and SD–Trp/–Leu/–His/–Ade/+Aba/+X-α-Gal denotes synthetically defined medium which lacks tryptophan, leucine, histidine, and adenine but contains Aureobasidin A and X-α-Gal. **(B)** HEK293 cells were co-transfected with Myc-PRMT5 and GFP or GFP-FAM47E constructs.

FAM47E in HEK293 cells. We observed that the overexpression of FAM47E did not affect the PRMT5–MEP50 interaction suggesting that FAM47E interacts with PRMT5 and MEP50 without affecting the PRMT5–MEP50 complex (Fig S2B). Collectively, the results obtained using different approaches, establish FAM47E as an interaction partner of PRMT5.

**FAM47E enhances the stability of PRMT5 protein**

Because PRMT5 interacts with FAM47E, we investigated if PRMT5 methylates FAM47E. For this, we overexpressed GFP-FAM47E and the known PRMT5 substrate, GFP-SmD3, as a positive control, in HEK293 cells in the presence and absence of PRMT5 inhibitor, EPZ015666 (Chan-Penebre et al, 2015). GFP-FAM47E and GFP-SmD3 proteins were immunoprecipitated from these cells and their methylation status was investigated using pan symmetric dimethyl arginine antibody (SYM10). We observed a strong methylation signal in SmD3 protein, and this signal was reduced in the SmD3 protein isolated from the cells treated with EPZ015666. However, we could not detect any methylation signal in FAM47E protein (Fig S3). These findings suggest that FAM47E is an interaction partner of PRMT5 and is unlikely to be its substrate.

Protein–protein interaction may result in alteration of (i) protein stability of the interacting proteins (ii) functional outcomes or (iii) both. We first tested if PRMT5-FAM47E interaction alters the stability of PRMT5 and/or FAM47E. To investigate this, we overexpressed GFP-tagged FAM47E and Myc-tagged PRMT5 individually or in combination. Whereas the overexpression of PRMT5 did not alter the levels of FAM47E, the overexpression FAM47E increased the PRMT5 protein levels (Fig S4). We next tested for the effect of FAM47E perturbation on the endogenous levels of PRMT5. For this, we depleted FAM47E levels by siRNA or overexpressed the GFP-tagged FAM47E in HEK293 cells and quantified the levels of PRMT5 protein in these cells by immunoblotting. We confirmed the knockdown of FAM47E by qRT-PCR (Fig S5) and immunoblotting (Fig 2B). We found that the overexpression of FAM47E increased the levels of PRMT5 protein by ~2.2-fold (Fig 2A) and the depletion of FAM47E reduced levels of PRMT5 protein significantly by ~39% (Fig 2B). This suggests that FAM47E regulates the steady state levels of PRMT5 in HEK293 cells.

FAM47E might regulate the levels of PRMT5 by (i) increasing the synthesis of PRMT5, (ii) stabilizing PRMT5 protein by binding to it and preventing its degradation or (iii) both. To gain mechanistic understanding on how FAM47E regulates PRMT5 protein levels, we first quantified the levels of PRMT5 transcripts in HEK293 cells

upon perturbation of FAM47E by using qRT-PCR. We observed that the knockdown of FAM47E did not alter the levels of PRMT5 mRNA levels significantly (Fig S6A). This suggests that the reduction of PRMT5 protein levels upon the depletion of FAM47E is not due to the decrease in the transcription of PRMT5. Interestingly, over-expression of FAM47E did not increase the levels of PRMT5 mRNA levels but decreased it by ~15% (Fig S6B). Contrarily, this suggests that the elevated levels of PRMT5 protein might reduce its own transcription through feedback inhibition. This could imply that the increase in the protein levels of PRMT5 upon overexpression of FAM47E could be due to increase in the protein stability of PRMT5. We hypothesized that FAM47E binding to PRMT5 enhances the stability of PRMT5 protein by inhibiting its proteasomal degradation. To test this, we quantified the PRMT5 levels in HEK293 cells upon overexpression or knockdown of FAM47E and treated with the proteasomal inhibitor, MG-132. We found that the treatment of MG-132 abolished the FAM47E dependent increase or decrease of PRMT5 levels suggesting that FAM47E inhibits the proteasome-mediated degradation of PRMT5 (Fig 2C and D). Taken together, these observations suggest that FAM47E interacts with PRMT5 and increases the stability of PRMT5 by preventing its proteasomal degradation.

Because the E3 ubiquitin ligase CHIP interacts with PRMT5 and promotes its proteasomal degradation through ubiquitination (Zhang et al, 2016), we investigated whether FAM47E–PRMT5 inter-action affects the binding of the E3 ubiquitin ligase CHIP with PRMT5. For this, we co-expressed GFP or GFP-PRMT5 and Myc-tagged CHIP with or without HA tagged FAM47E. We observed that the over-expression of FAM47E did not affect the PRMT5–CHIP interaction but on the contrary the overexpression of FAM47E enhanced PRMT5–CHIP interaction mildly (Fig S7). The mild enhancement of PRMT5–CHIP interaction might be due to the increase in the protein levels of PRMT5 upon overexpression of FAM47E. This suggests that the sta-bilization of PRMT5 by FAM47E is not mediated by disrupting the PRMT5–CHIP interaction. However, this does not rule out the pos-sibility that FAM47E–PRMT5 interaction might block the CHIP-mediated polyubiquitination of PRMT5. Because PRMT5 is ubiquiti-nated at multiple lysine residues (Zhang et al, 2016), it is also possible that FAM47E-PRMT5 interaction might inhibit the polyubiquitination of PRMT5 mediated by as yet unknown E3 ubiquitin ligase(s) that targets PRMT5. These findings lay foundation for future investigations to delineate the mechanisms underlying FAM47E inhibition of the proteasomal degradation of PRMT5.

Co-immunoprecipitation (Co-IP) was performed using GFP-Trap and the bound fractions were probed with Myc antibody. About 0.5% of the whole cell lysates which were used in Co-IP were probed with Myc antibody or GFP antibody. **(C)** HEK293 cells were co-transfected with HA-FAM47E and GFP or GFP-PRMT5 constructs. Co-IP was performed using GFP-Trap and the bound fractions were probed with HA antibody. About 1% of the whole cell lysates which were used in Co-IP were probed with HA antibody or GFP antibody. **(D)** Endogenous PRMT5 interacts with GFP tagged FAM47E. HEK293 cells were transfected with GFP or GFP-FAM47E constructs. Co-IP was performed using GFP-Trap and the bound fractions were probed with PRMT5 antibody. About 2% of the whole cell lysates which were used in immunoprecipitation were probed with PRMT5 antibody or GFP antibody. **(E)** PRMT5 interacts with FAM47E directly. GST pull-down assay was carried out using recombinant GST-FAM47E and His-RMT5 proteins. The bound fractions were probed with His antibody. About 4% of His-PRMT5 (middle blot) and 2% of GST-FAM47E (lower blot), which were used in the pull-down assay, were resolved in SDS–PAGE and stained with coomassie blue dye. **(F)** PRMT5 interacts with FAM47E at their endogenous levels in HEK293 cells. The cell lysates were prepared from HEK293 cells and immunoprecipitations were performed using rabbit IgG or FAM47E antibody and the bound fractions were probed with PRMT5 antibody. About 2% of the whole cell lysates which were used in immunoprecipitation were probed with PRMT5 antibody or FAM47E antibody. **(G)** The cell lysates were prepared from HEK293 cells and immunoprecipitations were performed using rabbit IgG or PRMT5 antibody and the bound fractions were probed with FAM47E antibody. About 0.1% of the whole cell lysates which were used in immunoprecipitation were probed with PRMT5 antibody or FAM47E antibody. Source data are available for this figure.

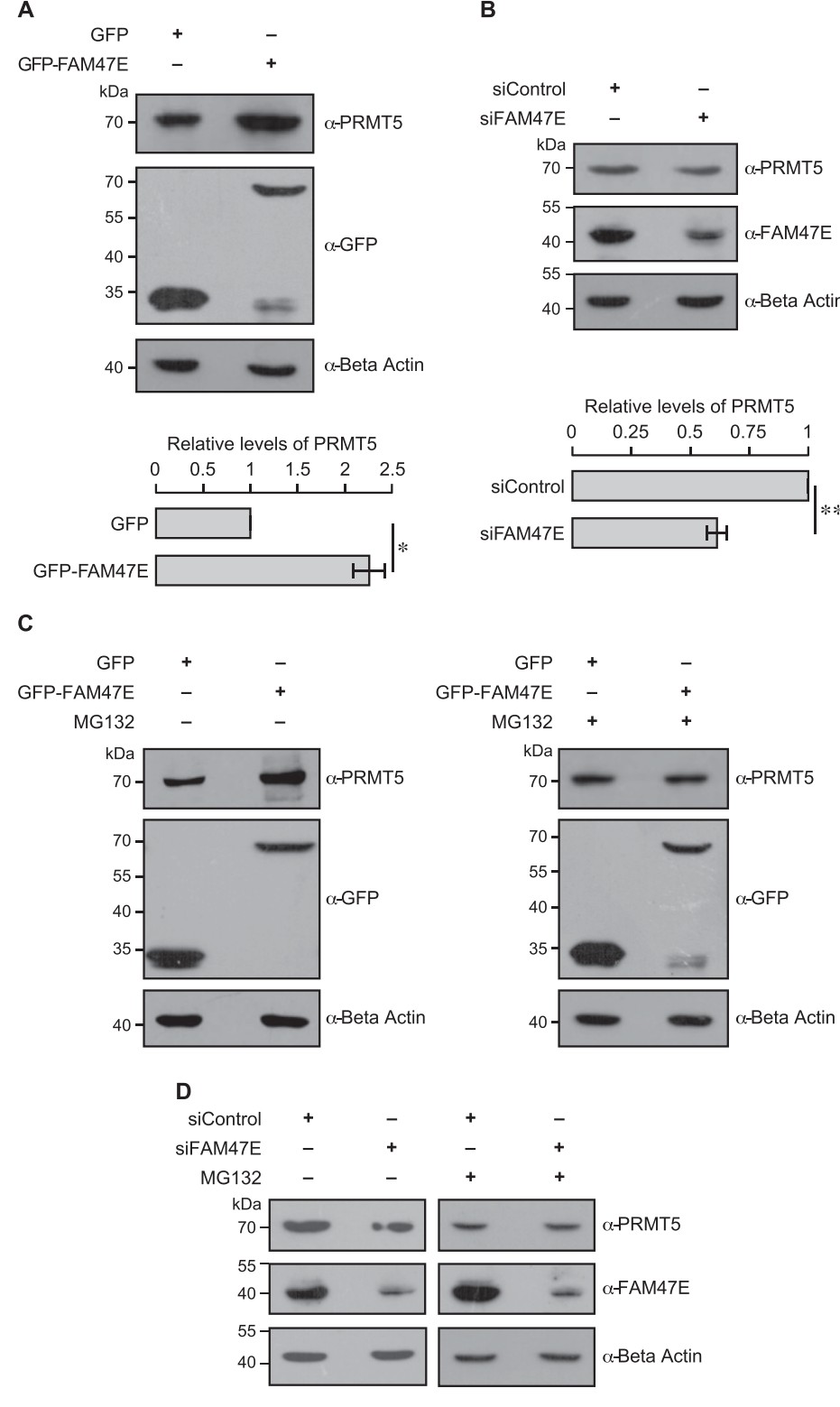

Figure 2. FAM47E increases the stability of PRMT5 protein.
**(A)** HEK293 cells were transfected with GFP vector or GFP-FAM47E construct. After 48 h of transfection, the cells were lysed, immunoblotted, and probed with PRMT5 antibody or GFP antibody or $\beta$ actin antibody (upper panel). The band intensities of PRMT5 and $\beta$ actin in the blots were quantified using ImageJ software and the relative ratios of PRMT5 signal to $\beta$ actin signal are plotted in the graph (lower panel). The values represent the mean of three independent experiments, with error bars representing SD. Statistical significance was assessed using two-tailed $t$ test. * indicates $P <$ 0.05. **(B)** HEK293 cells were transfected with control siRNA vector or FAM47E siRNA. After 48 h of transfection, the cells were lysed, immunoblotted, and probed with PRMT5 antibody or FAM47E antibody or $\beta$ actin antibody (upper panel). The band intensities of PRMT5 and $\beta$ actin in the blots were quantified using ImageJ software and the relative ratios of PRMT5 signal to $\beta$ actin signal are plotted in the graph (lower panel). The values represent the mean of three independent experiments, with error bars representing SD. Statistical significance was assessed using two-tailed $t$ test. ** indicates $P <$ 0.01. **(C)** HEK293 cells were transfected with GFP vector and GFP-FAM47E construct. After 40 h of transfection, the cells were treated with DMSO or MG-132 and incubated for 8 h. The cells were lysed, immunoblotted and probed with PRMT5 antibody or GFP antibody or $\beta$ actin antibody. **(D)** HEK293 cells were transfected with control siRNA vector or FAM47E siRNA. After 40 h of transfection, the cells were treated with DMSO or MG-132 and incubated for 8 h. The cells were lysed, immunoblotted, and probed with PRMT5 antibody or FAM47E antibody or $\beta$ actin antibody.

### FAM47E promotes the chromatin association of PRMT5 and histone arginine methylation

We next investigated whether FAM47E binding to PRMT5 affects the functionality of PRMT5. One of the well-established functions of PRMT5 is the epigenetic regulation of gene expression through histone arginine methylation. In this regard, we first tested if FAM47E binding affects the association of PRMT5 to chromatin. For this, we perturbed the FAM47E levels in HEK293 cells (Fig 3A) and isolated the soluble and chromatin fraction from these cells. We quantified the levels of PRMT5 in these fractions by immuno-blotting. We found that the overexpression of FAM47E decreased the levels of PRMT5 in soluble fractions by ~21% (Fig 3B, left panel) and increased the abundance of PRMT5 in chromatin fractions dramatically by ~2.2-fold (Fig 3C, left panel) suggesting that FAM47E–PRMT5 interaction profoundly increases the association of PRMT5 to the chromatin. We observed that the knockdown of FAM47E increased the PRMT5 levels mildly in soluble fractions by ~12% (Fig 3B, right panel) and decreased the PRMT5 level in chromatin fractions strongly by ~33% (Fig 3C, right panel). These findings imply that FAM47E–PRMT5 interaction not only stabilizes PRMT5 but also enhances its association with the chromatin.

We next investigated if the increase in chromatin association of PRMT5 upon the overexpression of FAM47E translates into increased methylation of histones by PRMT5. To test this, we quantified the histone modifications that could be introduced by PRMT5 viz. (i) symmetric dimethylation of arginine 2 of histone 3 (H3R2me2s), (ii) symmetric dimethylation of arginine 8 of histone 3 (H3R8me2s), and (iii) symmetric dimethylation of arginine 3 of histone 4 (H4R3me2s) in histones isolated from HEK293 cells, upon overexpression or knockdown of FAM47E. We observed that the overexpression of FAM47E significantly increases the levels of H3R2me2s, H3R8me2s, and H4R3me2s modifications (~2.1-folds; Fig 4A and C). Similarly, the depletion of FAM47E decreased the levels of PRMT5 mediated histone arginine methylation modifications by ~40% (Fig 4B and D). These findings provide the functional significance of our observation that FAM47E regulates chromatin association of PRMT5 (Fig 3). Collectively, these observations suggest that (i) FAM47E is essential for the physiological epigenetic functions of PRMT5 and (ii) dysregulated levels of FAM47E might result in PRMT5-mediated detrimental effects.

FAM47E might reinforce the PRMT5-mediated epigenetic control of gene expression through histone arginine methylation. To investigate this, we perturbed the levels of FAM47E in HEK293 cells through overexpression or knockdown (Figs 5A and S8A) and quantified the expression of few well known PRMT5 target genes (Mongiardi et al, 2015; Sohail & Xie, 2015) by quantitative PCR. We observed that overexpression of FAM47E significantly (i) reduces the expression of the carbamoyl-phosphate synthetase 2, aspartate transcarbamylase, and dihydroorotase (CAD) and cyclin D1 (CCND1) genes (Fig 5B), which are negatively regulated by PRMT5 (Mongiardi et al, 2015) and (ii) increases the expression of the doublecortin-like kinase 1 (DCLK1), pentraxin-related protein (PTX3), and tumor necrosis factor α–induced protein 3 (TNFAIP3) genes (Fig 5B), which are positively regulated by PRMT5 (Sohail & Xie, 2015). These results suggest that FAM47E overexpression mimics the PRMT5 overexpression in terms of epigenetic regulation of PRMT5-target genes.

To ensure that the effect of FAM47E overexpression on the regulation of PRMT5-target genes expression is mediated by PRMT5, we overexpressed FAM47E in PRMT5-depleted HEK293 cells and quantified the expression of PRMT5 target genes by qRT-PCR. The knockdown of PRMT5 in these cells was confirmed by qRT-PCR and immunoblotting (Figs 5A and S8B). We observed that the effect of FAM47E overexpression on the expression of PRMT5 target genes is lost in most cases or reversed in a few, in PRMT5-depleted cells, suggesting that FAM47E regulates the PRMT5 target genes expression through PRMT5 (Fig 5B). When FAM47E was depleted, there was an increase in the expression of the genes (CAD and CCND1) that are negatively regulated by PRMT5 and reduced expression of genes (DCLK1, PTX3, and TNFAIP3) that are known to be positively regulated by PRMT5, suggesting that FAM47E facilitates PRMT5-mediated epigenetic regulation of gene expression (Fig 5B).

Because FAM47E regulates the PRMT5-mediated epigenetic regulation, we investigated the binding of FAM47E and PRMT5 at the promoters of the tested PRMT5 target genes by using chromatin immunoprecipitation (ChIP). The details about the promoter regions of these genes for ChIP analyses were obtained from previous studies (Khan et al, 2007; Oconnell et al, 2015; Rubino et al, 2017; Hermosilla et al, 2018; Lee et al, 2019). We found that both FAM47E and PRMT5 proteins were enriched at the promoter regions of the tested PRMT5 target genes. This indicates that FAM47E binds to the promoters of the PRMT5 target genes along with PRMT5 and contributes to PRMT5-mediated epigenetic regulation (Fig 5C). Taken together, these results establish that FAM47E tunes the PRMT5-mediated epigenetic regulation of gene-expression by enhancing the association of PRMT5 with chromatin and subsequent histone arginine methylation modifications. Because FAM47E is distributed both in the cytoplasm and nucleus (Fig S9), we investigated whether FAM47E affects the non-chromatin functions of PRMT5. To address this, we analyzed the methylation levels of SmD3, a well-studied non-chromatin substrate of PRMT5 (Friesen et al, 2001; Meister et al, 2001) upon overexpression of FAM47E. For this, we overexpressed GFP-SmD3 with or without FAM47E-HA in HEK293 cells. GFP-SmD3 was immunoprecipitated from these cells and its methylation levels were assessed using pan symmetric dimethyl arginine antibody, SYM10. The reliability of SYM10 antibody in detecting PRMT5-mediated methylation of SmD3 was confirmed by analyzing the methylation levels of SmD3 in the cells which were treated with or without PRMT5 inhibitor, EPZ015666 (Chan-Penebre et al, 2015) (Fig S3). We found that the overexpression of FAM47E did not alter methylation levels of SmD3 suggesting that FAM47E primarily affects the chromatin-associated epigenetic functions of PRMT5 (Fig S10). Nevertheless, this does not rule out the plausible role of FAM47E in affecting other non-chromatin functions of PRMT5. Further detailed investigations are required to unravel this aspect.

### FAM47E promotes cell proliferation and clonogenic capacity of the cells via PRMT5 axis

PRMT5 levels are elevated in several cancer types and are associated with poor clinical outcomes (Bao et al, 2013; Győrffy et al, 2013; Stopa et al, 2015; Xiao et al, 2019). PRMT5 overexpression increases cell proliferation and colony forming capacity of the cells and the knockdown of PRMT5 reduces the cell proliferation and

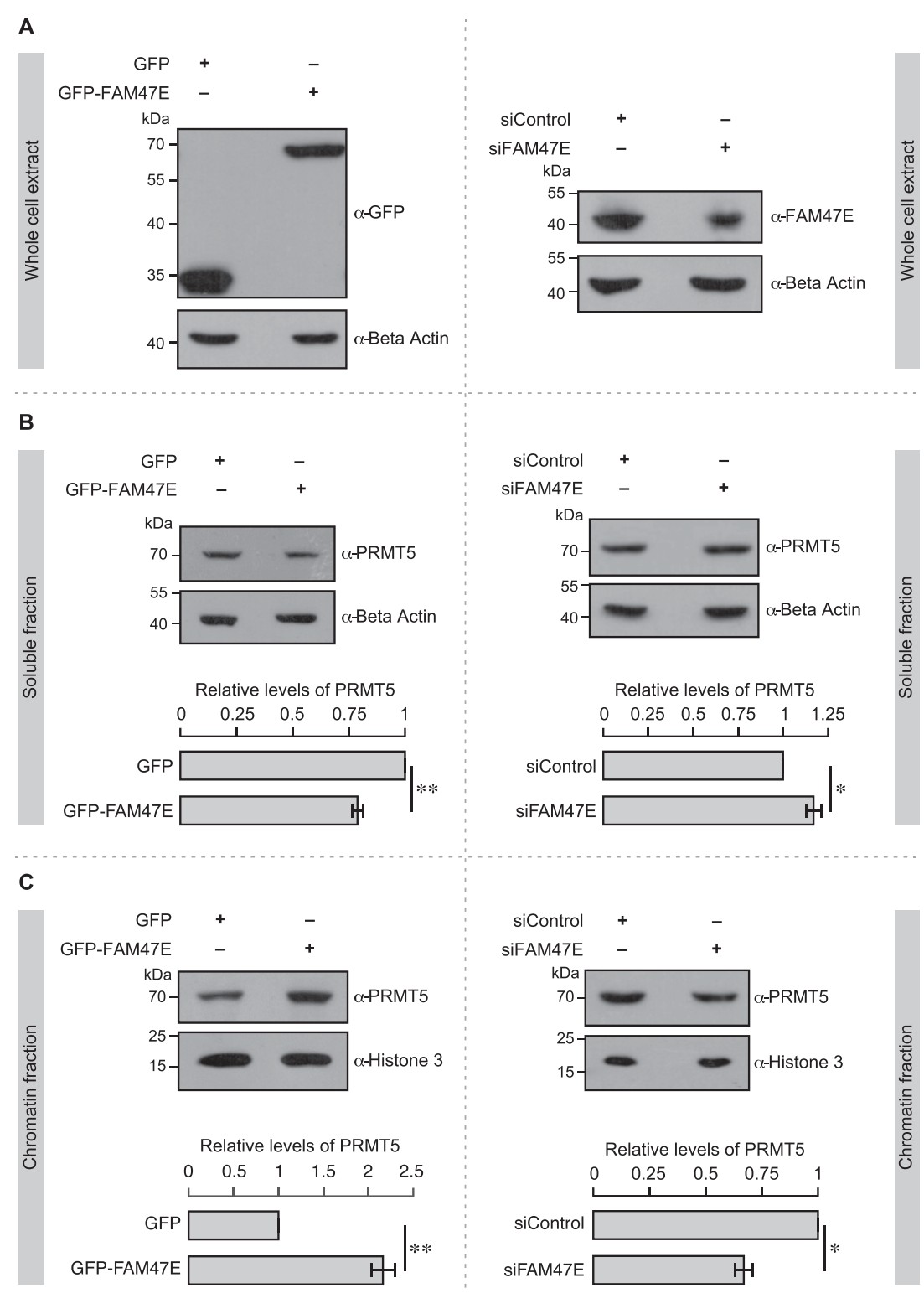

**Figure 3.  FAM47E enhances the chromatin association of PRMT5.**
**(A)** HEK293 cells were transfected with, GFP vector or GFP-FAM47E construct or control siRNA or FAM47E siRNA. After 48 h of transfection, the cells were lysed, immunoblotted, and probed with GFP antibody or $\beta$ actin antibody (left panel) and FAM47E antibody or $\beta$ actin antibody (right panel). **(B)** HEK293 cells were transfected with GFP vector or GFP-FAM47E construct or control siRNA or FAM47E siRNA. After 48 h of transfection, the cells were lysed, the soluble fractions of the nuclei were isolated, and immunoblotting was performed using PRMT5 antibody or $\beta$ actin antibody (upper panels). The band intensities of PRMT5 and $\beta$ actin in the blots were quantified using ImageJ software and the relative ratios of PRMT5 signal to $\beta$ actin signal are plotted in the graph (lower panels). The values represent the mean of three independent

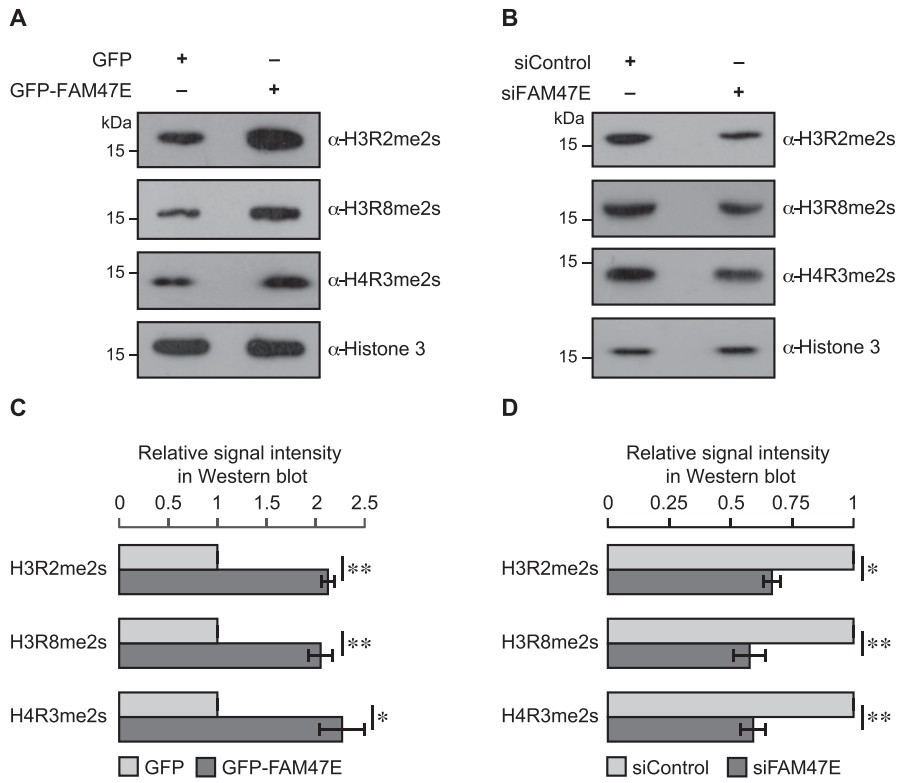

**Figure 4.  FAM47E promotes the histone methylation activity of PRMT5.**
**(A, B)** HEK293 cells were transfected with GFP vector or GFP-FAM47E construct (A) or control siRNA or FAM47E siRNA (B). The cells were harvested after 48 h of transfection and histones were isolated. The isolated histones were immunoblotted and probed with H3R2me2s antibody or H3R8me2s antibody or H4R3me2s antibody or histone 3 antibody. **(C, D)** The relative ratios of histone arginine methylation signals to histone 3 signal are plotted in the graph for FAM47E overexpression (C) and knockdown (D) conditions. The band intensities of histone arginine methylation modifications and histone 3 in the blots were quantified using ImageJ software. The values represent the mean of three independent experiments, with error bars representing standard deviations. Statistical significance was assessed using two-tailed *t* test. * indicates *P* < 0.05 and ** indicates *P* < 0.01.

colony forming capacity of the cells (Pal et al, 2004; Scoumanne et al, 2009; Wei et al, 2012; Stopa et al, 2015). Because FAM47E increases the PRMT5 protein levels and its chromatin association, we investigated the effect of FAM47E overexpression or knock down (Fig 6A) on cell proliferation and colony forming capacity of HeLa cells. The knock down of FAM47E in HeLa cells was confirmed by qRT-PCR and immunoblotting (Figs 6A and S11A). The effect of FAM47E perturbation on cell proliferation was investigated by cell counting and MTT assays. We found that overexpression of FAM47E increased the cell proliferation and the depletion of FAM47E decreased the cell proliferation as reflected by the decrease or increase in doubling time, respectively (Figs 6B and S12). We observed similar results in the MTT assay as well (Fig 6C). We next probed the effect of FAM47E perturbation on the colony forming capacity of the HeLa cells. We observed that colony forming capacity of the cells increased by ~45% upon the overexpression of FAM47E and decreased by ~29% upon the knockdown of FAM47E (Fig 6D). These findings suggest that elevated levels of FAM47E can have oncogenic potential.

Based on our above observations, we hypothesized that increased cell proliferation and colony forming capacity upon FAM47E overexpression could be mediated through increase in PRMT5

levels/activity. To test if the effect of FAM47E overexpression on cell proliferation and clonogenic capacity is mediated by PRMT5, we overexpressed the FAM47E in PRMT5-depleted HeLa cells and investigated the cell proliferation and clonogenic capacity of the cells. The knockdown of PRMT5 in these cells was confirmed by qRT-PCR and immunoblotting (Figs 6A and S11B). We observed that the effect of FAM47E overexpression on the cell proliferation and colony forming capacity is either lost or reduced in PRMT5-depleted cells suggesting that FAM47E increases the cell proliferation and colony forming capacity of the cells via PRMT5 axis (Fig 6B–D). Taken together, these data suggest that the FAM47E is important for cell proliferation mediated by PRMT5 and when dysregulated could have oncogenic potential.

# Discussion

Genome wide association studies indicated that the FAM47E is associated with chronic kidney disease and Parkinson's disease (Ledo et al, 2015; Blauwendraat et al, 2019). However, nothing is known about the interaction partner or the function(s) of this protein. Here, we report that FAM47E interacts and regulates the

experiments, with error bars representing standard deviations. Statistical significance was assessed using two-tailed *t* test. * indicates *P* < 0.05 and ** indicates *P* < 0.01.
**(C)** HEK293 cells were transfected with GFP vector or GFP-FAM47E construct or control siRNA or FAM47E siRNA. After 48 h of transfection, the cells were lysed, the chromatin fractions were prepared by benzonase digestion and immunoblotting was performed using PRMT5 antibody or histone 3 antibody (upper panels). The band intensities of PRMT5 and histone 3 in the blots were quantified using ImageJ software and the relative ratios of PRMT5 signal to histone 3 signal are plotted in the graph (lower panels). The values represent the mean of three independent experiments, with error bars representing standard deviations. Statistical significance was assessed using two-tailed *t* test. * indicates *P* < 0.05 and ** indicates *P* < 0.01.

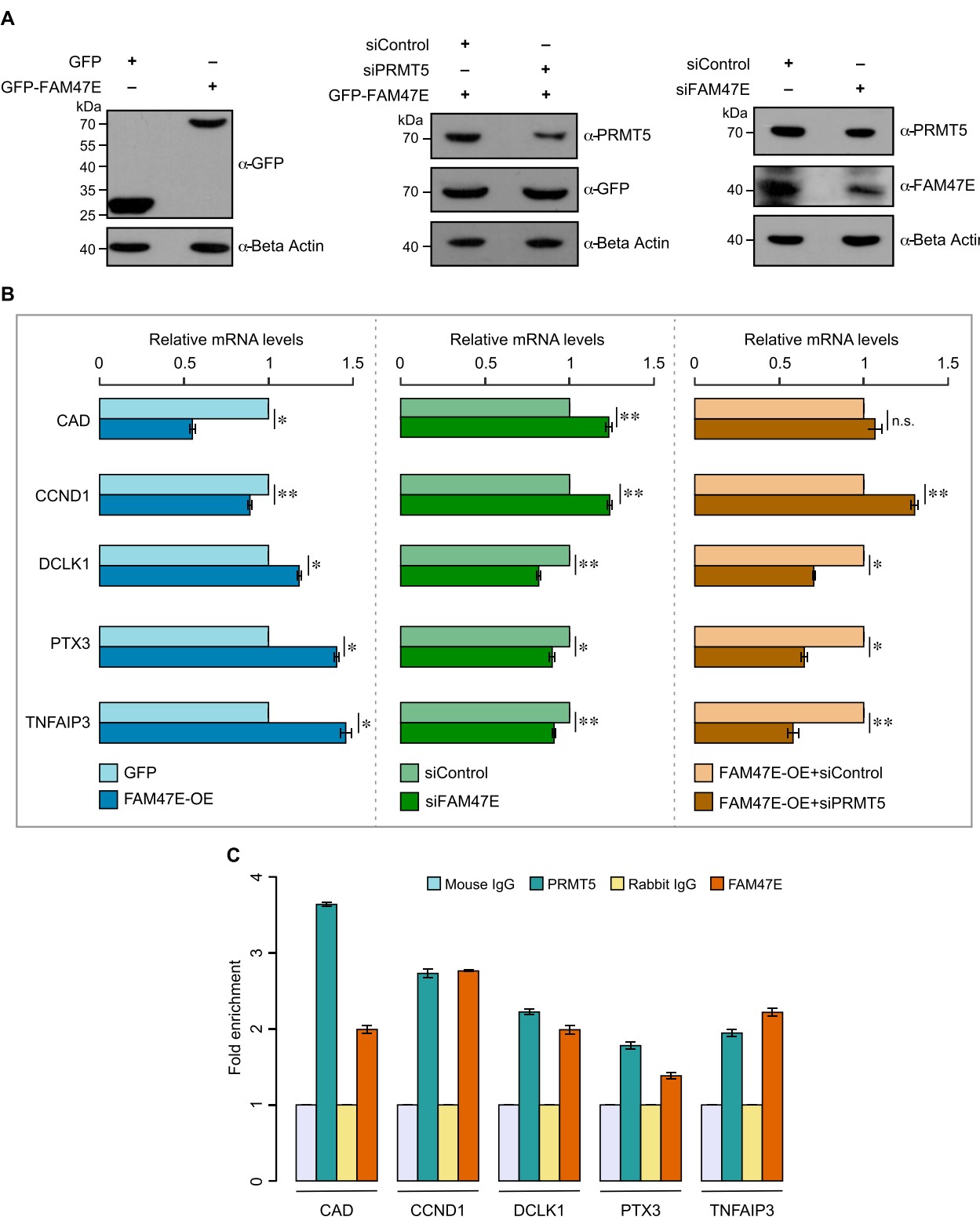

**Figure 5. FAM47E regulates the expression of PRMT5 target genes.**
**(A, B)** HEK293 cells were transfected with GFP vector or GFP-FAM47E construct or control siRNA or FAM47E siRNA or co-transfected with GFP-FAM47E construct and control siRNA or PRMT5 siRNA. After 48 h of transfections, the whole cell lysate and total RNA were isolated from these cells. **(A)** The cell lysates were immunoblotted and probed with GFP antibody or PRMT5 antibody or FAM47E antibody or β actin antibody. **(B)** The transcripts levels of the indicated PRMT5 target genes in these cells were quantified by using quantitative RT-PCR. The mRNA levels of indicated PRMT5 target genes were normalized to GAPDH expression and are presented relative to the control sample. The values represent the mean of three independent experiments, with error bars representing standard deviations. Statistical significance was assessed using two-tailed *t* test. * indicates *P* < 0.05, ** indicates *P* < 0.01, and n.s. indicates not significant. **(C)** The chromatin was prepared from HEK293 cells and the ChIP was performed using mouse IgG or PRMT5 antibody or rabbit IgG or FAM47E antibody. The association of PRMT5 and FAM47E with the promoters of PRMT5 target genes was investigated by analyzing the immunoprecipitated DNA using quantitative RT-PCR. Data are presented as fold enrichment relative to the control IgG binding and the values represent the mean of three independent experiments, with error bars representing standard deviations.

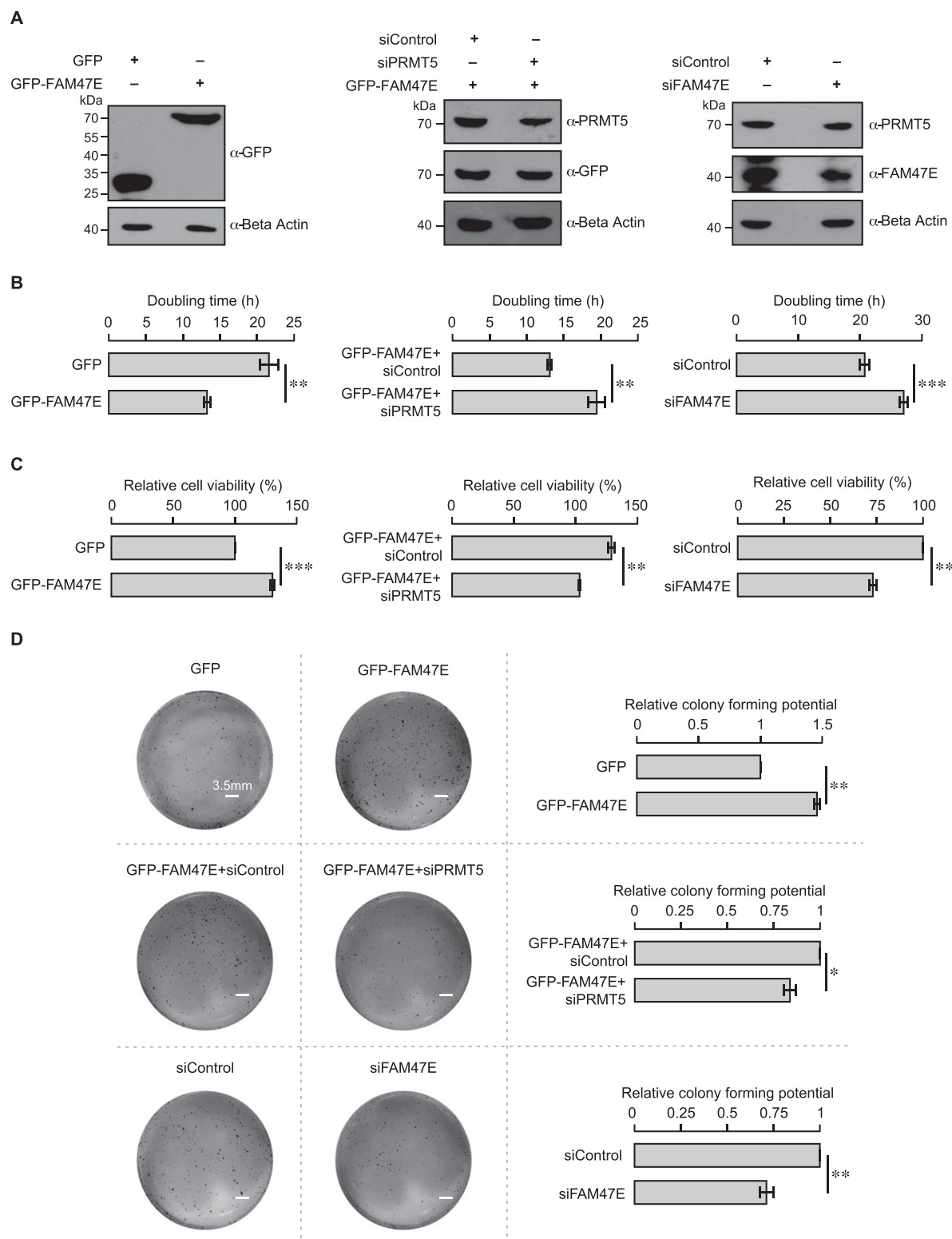

**Figure 6. FAM47E increases cell proliferation and the clonogenic potential of HeLa cells through PRMT5.**
**(A)** HeLa cells were transfected with GFP vector or GFP-FAM47E construct or control siRNA or FAM47E siRNA or co-transfected with GFP-FAM47E construct and control siRNA or PRMT5 siRNA. After 48 h of transfection, the cells were lysed and probed with GFP antibody or PRMT5 antibody or FAM47E antibody or β actin antibody. **(B)** HeLa cells were transfected with GFP vector or GFP-FAM47E construct or control siRNA or FAM47E siRNA or co-transfected with GFP-FAM47E construct and control siRNA or PRMT5 siRNA. The cells were counted after 48, 72 and 96 h of post-transfection (Fig S12) and the doubling times were calculated. The values in the graph represent the mean of three independent experiments, with error bars representing SD. Statistical significance was assessed using two-tailed *t* test. ** indicates *P* < 0.01 and *** indicates

functions of the versatile arginine methyltransferase PRMT5 (Fig 7). These findings provide the first insights into the functional role(s) of FAM47E. On the other hand, several interaction partners of PRMT5 and their functional outcomes have been extensively studied. Specifically, MEP50 forms an octameric complex with PRMT5 and regulates its enzymatic activity and its levels (Friesen et al, 2002; Gonsalvez et al, 2007; Antonysamy et al, 2012; Ho et al, 2013; Saha & Eckert, 2015; Saha et al, 2016; Chen et al, 2017). Strikingly, here we report that FAM47E increases the stability of PRMT5 and enhances its chromatin methylation activity (Fig 7). Although both the proteins seem to interact with PRMT5 and have overlapping effects, these could represent two distinct modes of regulation and function for PRMT5. For instance, here we show that FAM47E affects the stability of PRMT5 by inhibiting its proteasomal degradation. However, the mechanisms by which MEP50 regulates the levels of PRMT5 is unknown. In terms of functional impact, FAM47E enhances the chromatin methylation activity of PRMT5, by increasing its association with chromatin. On the other hand, MEP50 enhances the enzymatic activity of PRMT5 by increasing its affinity towards the substrate and the cofactor (Antonysamy et al, 2012). Further research is required to delineate the molecular mechanisms underlying the regulation and functional outcomes of PRMT5 upon binding with FAM47E and MEP50.

PRMT5 is a versatile protein which is involved in (i) epigenetic regulation via chromatin modifications and (ii) regulation of various other cellular processes through methylation and interaction of non-histone proteins. We found that the FAM47E increases PRMT5-mediated chromatin modifications by enhancing its association to the chromatin strongly and by decreasing the levels of PRMT5 in soluble fractions. This suggests that FAM47E promotes the epigenetic functions of PRMT5 and mitigates its non-epigenetic functions.

We also report that elevated levels of FAM47E could contribute the oncogenic properties of the cells as it increases the cell proliferation and colony forming capacity of the cells and demonstrate that the oncogenic functions of FAM47E is mediated by PRMT5. PRMT5 levels are up-regulated in several cancers and the depletion of PRMT5 reduces the carcinogenic properties of cells which makes the PRMT5 enzyme as an important therapeutic target for cancer therapy. This led to the discovery of selective inhibitors for PRMT5 enzyme and some of them entered clinical trials (Chan-Penebre et al, 2015; Bonday et al, 2018; Gerhart et al, 2018; Lin & Luengo, 2019; Zhou et al, 2019). However, most of these inhibitors target the enzymatic activity of PRMT5 which affects wide range of functions that are mediated by PRMT5 in the cell. Our data demonstrate that FAM47E-PRMT5 interaction promotes the stability and epigenetic regulations of PRMT5. Interestingly, in conditions, especially in esophageal cancer, where PRMT5 is not up-regulated but FAM47E was significantly up-regulated, substantial proportion of targets

positively regulated by PRMT5 were also significantly up-regulated (Fig S13). This suggests that FAM47E-mediated regulation of PRMT5 targets through stabilization of PRMT5 protein levels and increased association to chromatin could facilitate the enhanced expression of PRMT5 targets. This implies that the disruption of this interaction by small molecular inhibitors might serve as an alternative strategy for the preferential inhibition of PRMT5 epigenetic functions, which can be exploited as a specific therapy for the cancers in which PRMT5 mediated epigenetic signaling is dysregulated.

## Materials and Methods

### Cloning, expression, and purification

Using cDNA prepared from HEK293 cells, full-length FAM47E (NM_001242936.1, Isoform 2) and PRMT5 (NM_006109.4) sequences were PCR-amplified and cloned in different vectors. FAM47E was cloned in pGADT7 vector (Clontech) using EcoRI and BamHI sites, pGEX-6P2 vector (GE Healthcare) using BamHI and XhoI sites, and pCDNA3-EGFP vector (Invitrogen) using BamHI and EcoRI sites to generate pGADT7-FAM47E, pGEX-FAM47E, and pCDNA-GFP-FAM47E constructs, respectively. The oligo encoding the HA tag was introduced in to pCDNA4/myc-HisA vector (Invitrogen) using XhoI and ApaI sites to generate pCDNA4-HA vector. FAM47E was also sub-cloned in pCDNA4-HA using BamHI and EcoRI sites to generate pCDNA4-HA-FAM47E construct. Similarly, full-length PRMT5 was cloned in pGBKT7 vector (Clontech) using EcoRI and BamHI sites, pCDN4/myc-HisA (Invitrogen) using EcoRI and XhoI sites, pET28a (Novagen) using BamHI and XhoI sites and pEGFP-C1 vector (Clontech) using XhoI and BamHI sites to generate pGBKT7-PRMT5, pCDNA4-Myc-PRMT5, pET28-PRMT5, and pEGFP-PRMT5 constructs, respectively. The sequence encoding the full length MEP50 (NM_024102.4) and the E3 ubiquitin ligase CHIP (NM_005861.4) was cloned in pCDN4/Myc-HisA vector using BamHI and XhoI sites to generate pCDNA4-Myc-MEP50 and pCDNA4-Myc-CHIP constructs, respectively. The sequence encoding the full-length SmD3 (NM_004175.5) was cloned in pEGFP-C1 vector using EcoRI and BamHI sites to generate pEGFP-SmD3 construct. The bacterial expression and purification of GST-tagged FAM47E and His-tagged PRMT5 was performed as described previously (Verma et al, 2017; Awasthi et al, 2018).

### Y2H screening

Yeast two hybrid (Y2H) screening was performed using Matchmaker Gold Yeast two Hybrid system (Clontech) as per the manufacturer's instructions. To screen the interaction partners for PRMT5, pGBKT7-PRMT5

---

P < 0.001. **(C)** HeLa cells were transfected with GFP vector or GFP-FAM47E construct or control siRNA or FAM47E siRNA or co-transfected with GFP-FAM47E construct and control siRNA or PRMT5 siRNA. After 48 h of transfections, MTT assay was carried out. The values in the graph represent the mean of three independent experiments, with error bars representing standard deviations. Statistical significance was assessed using two-tailed *t* test. ** indicates P < 0.01 and *** indicates P < 0.001. **(D)** HeLa cells were transfected with GFP vector or GFP-FAM47E construct or control siRNA or FAM47E siRNA or co-transfected with GFP-FAM47E construct and control siRNA or PRMT5 siRNA. The colony-forming capacities of these cells were analyzed by staining the cells with crystal violet after 10 d of transfection. Experiments were performed in triplicates and the representative images are provided in the left panel. The scale bar is depicted. The colony numbers were counted using ImageJ software. The values in the graph represent the mean of three independent experiments, with error bars representing SD (right panel). Statistical significance was assessed using two-tailed *t* test. * indicates P < 0.05 and ** indicates P < 0.01.

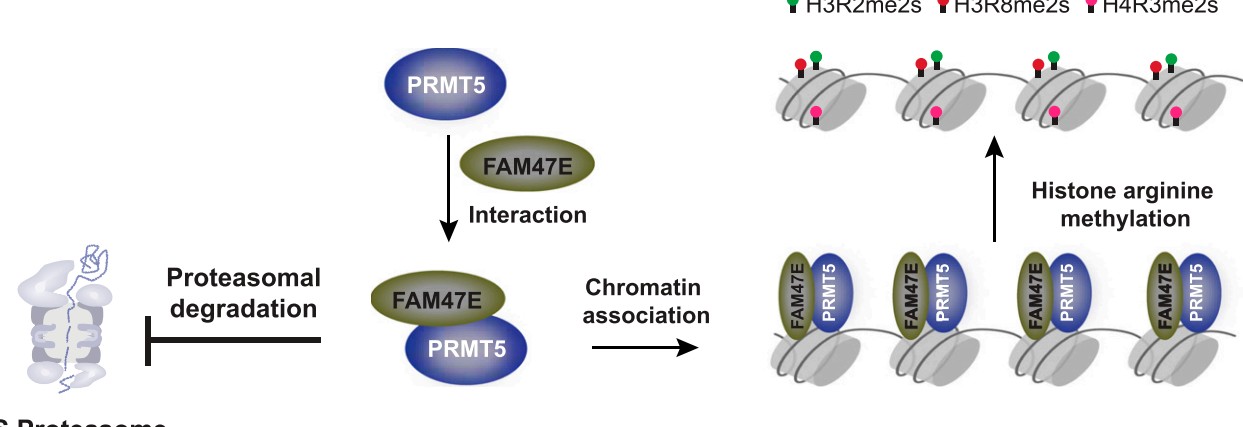

**H3R2me2s**  **H3R8me2s**  **H4R3me2s**

PRMT5

FAM47E

**Interaction**

**Proteasomal degradation**

FAM47E
PRMT5

**Chromatin association**

**20S Proteasome**

**Histone arginine methylation**

FAM47E PRMT5    FAM47E PRMT5    FAM47E PRMT5    FAM47E PRMT5

**Figure 7. Schema representing the regulatory role of FAM47E upon binding with PRMT5.**

construct was used as a bait and human normalized cDNA library (Cat. no. 630480; Clontech) were used as prey. The normalized human cDNA library lacks the abundantly expressed transcripts, resulting in improved representation of low abundance cDNA. The positive interactions in Y2H screening were identified by profiling the expression of reporter genes. The screening identified FAM47E as putative interaction partner for PRMT5.

The authenticity of the PRMT5–FAM47E interaction was investigated in Y2H assay by co-transforming pGBKT7-PRMT5 with pGADT7 vector or pGADT7-FAM47E construct in MM Gold yeast strain (Clontech). Similarly, co-transformation was also carried out using pGADT7-FAM47E construct and pGBKT7 vector. The transformants were plated in a (i) synthetically defined medium which lacks tryptophan and leucine (SD–Trp/–Leu), (ii) synthetically defined medium which lacks tryptophan and leucine but contains Aureobasidin A and X-α-Gal (SD–Trp/–Leu/+Aba/+X-α-Gal) and (iii) synthetically defined medium which lacks tryptophan, leucine, histidine, and adenine but contains Aureobasidin A and X-α-Gal (SD–Trp/–Leu/–His/–Ade/+Aba/+X-α-Gal) (Clontech). SD–Trp/–Leu medium allows the growth of all the co-transformants, whereas SD–Trp/–Leu/+Aba/+X-α-Gal media allows the growth of co-transformants which express the two reporter genes viz. *AUR1-C* and *MEL1*. Contrarily, SD–Trp/–Leu/–His/–Ade/+Aba/+X-α-Gal medium selects the co-transformants that express the four reporter genes viz. *HIS3, ADE2, AUR1-C*, and *MEL1*. The expression of *AUR1-C* confers resistance to the toxic drug, Aureobasidin A, and the *MEL1* codes for an enzyme, α-galactosidase which acts on the chromogenic substrate X-α-Gal. The positive interactions were confirmed by assessing the expression of reporter genes.

### Cell-culture and transfection

HEK293 and HeLa cells were purchased from National Centre for Cell Science, India and grown in DMEM (HiMedia) supplemented with 5% fetal bovine serum (HiMedia) and glutamine–penicillin–streptomycin solution (HiMedia). The cells were grown in incubator supplied with 5% $CO_2$. The plasmid constructs were transfected using standard calcium phosphate precipitation method. The

siRNAs, siControl (siRNA Negative control, Cat. no. SR-CL000-005; Eurogentec), siFAM47E (SMARTpool - siGENOME siRNA targeting FAM47E, Cat. no. M185579-00-0005; Dharmacon), and siPRMT5 (5′-CUU UGA GAC UGU GCU UUA U 3′) were transfected using Lipofectamine 2000 Transfection Reagent (Thermo Fisher Scientific).

### Co-immunoprecipitation

Co-Immunoprecipitation (Co-IP) experiments were performed to investigate the interaction of PRMT5 and FAM47E. For forward Co-IP, pCDNA4-Myc-PRMT5 construct was co-transfected with either pCDNA-GFP vector or pCDNA-GFP-FAM47E construct in HEK293 cells. For reverse Co-IP, pCDNA4-HA-FAM47E construct was co-transfected with either pEGFP vector or pEGFP-PRMT5 construct in HEK293 cells. After 48 h of transfection, the cells were harvested and lysed in RIPA buffer (50 mM Tris [pH 8.0], 150 mM NaCl, 0.5 mM EDTA, 0.1% NP40, 0.1% SDS, 0.5% deoxycholate, and protease inhibitor cocktail [Roche]). For co-immunoprecipitation, the cell lysates were incubated with GFP-Trap A beads (ChromoTek) for 8 h at 4°C. After the incubation, the beads were washed extensively with RIPA buffer.

To investigate the interaction of FAM47E and MEP50, pCDNA-Myc-MEP50 construct was co-transfected with either pCDNA-GFP vector or pCDNA-GFP-FAM47E construct in HEK293 cells. To study the influence of FAM47E on PRMT5-MEP50 interaction, pCDNA-Myc-MEP50 construct was co-transfected with the combinations of pEGFP vector and pCDNA-HA vector or pEGFP-PRMT5 construct and pCDNA-HA vector or pEGFP-PRMT5 construct and pCDNA-HA-FAM47E construct in HEK293 cells. To study the influence of FAM47E on PRMT5-CHIP interaction, pCDNA-Myc-CHIP construct was co-transfected with the combinations of pEGFP vector and pCDNA-HA vector or pEGFP-PRMT5 construct and pCDNA-HA vector or pEGFP-PRMT5 construct and pCDNA-HA-FAM47E construct in HEK293 cells. After 48 h of transfection, the cells were harvested and lysed in lysis buffer (10 mM Tris, pH:7.5, 150 mM NaCl, 0.5 mM EDTA, 0.5% NP-40 and Protease Inhibitor Cocktail [Roche]). For co-immunoprecipitation, the cell lysates were incubated with GFP-Trap A beads (ChromoTek) for 8 h at 4°C. After the incubation, the

beads were washed extensively with lysis buffer or wash buffer (10 mM Tris, pH:7.5, 300 mM NaCl, 0.5 mM EDTA, 0.3% NP-40, and protease inhibitor cocktail [Roche]).

The bound proteins were eluted by boiling the beads at 100°C for 5 min, and the eluted proteins were separated in 12% SDS–PAGE and transferred to PVDF membrane (GE Healthcare). The membrane was blocked overnight with blocking agent (GE Healthcare), washed with TTBS buffer (25 mM Tris, 150 mM NaCl, and 0.1% Tween 20) and probed with anti-Myc antibody (Cat. no. sc-40; Santa Cruz Biotechnology) or anti-HA antibody (Cat. no. 11867423001; Roche). The whole cell extract which was used in co-immunoprecipitation were immunoblotted and probed with anti-Myc antibody or anti-HA antibody or anti-GFP antibody (Cat. no. 632375; Clontech). The blots were developed using Super signal West Pico chemiluminescent substrate (Thermo Fisher Scientific) as per the manufacturer's instructions and the images were captured in X-ray sheets in darkroom.

To investigate the interaction of GFP-tagged FAM47E with endogenous PRMT5, HEK293 cells were transfected with either pCDNA-GFP vector or pCDNA-GFP-FAM47E construct. After 48 h of transfection the cells were lysed in RIPA buffer and immunporeciptation was carried out using GFP-Trap A beads (ChromoTek) as described above. The bound fractions were immunoblotted and probed with anti-PRMT5 antibody (Cat no. 07-405; Merck Millipore).

To investigate the interaction PRMT5 and FAM47E at their endogenous levels, HEK293 cells were lysed in a lysis buffer (10 mM Tris [pH 7.5], 150 mM NaCl, 0.5 mM EDTA, 0.5% NP40, and protease inhibitor cocktail). The lysates were incubated with rabbit IgG antibody (Cat. no. 2729; Cell Signaling Technology) or anti-FAM47E antibody (Cat. no. PA5-46681; Thermo Fisher Scientific) or anti-PRMT5 antibody for 1 h at 4°C. After the incubation, 30 $\mu$l of Protein A Dynabeads (Invitrogen) were added to the lysates and incubated in rotator for 10 h at 4°C. The beads were washed thrice with wash buffer (10 mM Tris [pH 7.5], 150 mM NaCl, 0.5 mM EDTA, and 0.1% NP40). The bound proteins were eluted by boiling the beads at 100°C for 5 min. The eluted proteins were immunoblotted and probed with anti-FAM47E primary antibody or anti-PRMT5 antibody. The whole cell extract which was used in immunoprecipitation was also immunoblotted and probed with anti-FAM47E primary antibody or anti-PRMT5 antibody.

### GST pull-down assay

Glutathione sepharose 4B (GE Healthcare) beads were coupled with either 50 $\mu$g of GST protein or GST-tagged FAM47E protein in ice cold interaction buffer containing 20 mM Hepes (pH 7.5), 150 mM KCl, 0.2 mM DTT, 1 mM EDTA, and 10% glycerol. Then the beads were blocked with interaction buffer containing 5% bovine serum albumin for 1 h at 4°C. The blocked beads were incubated with 25 $\mu$g of His-tagged PRMT5 protein in binding buffer (10 mM Tris, 150 mM NaCl, 0.5 mM EDTA, and 0.1% NP40) for 3 h at 4°C. After incubation the beads were washed thrice with wash buffer (10 mM Tris, 300 mM NaCl, 0.5 mM EDTA, and 0.5% NP40). The bound proteins were eluted by boiling the beads with 2× LAP at 100°C for 5 min. The eluted proteins were immunoblotted and probed with anti-His antibody (Cat. no. A00186-100; GenScript).

### PRMT5 stability assay

To investigate the stability of Myc-PRMT5 protein upon GFP-FAM47E overexpression or vice versa, HEK293 cells were transfected with pCDNA-GFP-FAM47E construct or pCDNA4-Myc-PRMT5 construct individually or in combination. The cells were collected 48 h of posttransfection and lysed in RIPA buffer. The lysates were immunoblotted and probed with anti-GFP antibody or anti-Myc antibody or anti-GAPDH antibody (Cat. no. MA5-15738; Thermo Fisher Scientific).

To study the effect of FAM47E perturbation on endogenous PRMT5 levels, HEK293 cells were transfected with control siRNA or FAM47E siRNA or pCDNA-GFP vector or pCDNA-GFP-FAM47E construct. After 40 h of transfection, the cells were treated with or without 50 $\mu$M of MG-132 and incubated for 8 h and lysed in RIPA buffer. The lysates were immunoblotted and probed with anti-PRMT5 antibody or anti-FAM47E antibody or anti-GFP antibody or anti-$\beta$ actin antibody (Cat. no. A2228; Sigma-Aldrich). The band intensities were quantified using the ImageJ software.

### Chromatin association studies

The levels of PRMT5 association with the chromatin was investigated as described previously (Bian et al, 2015) with few modifications. Briefly, HEK293 cells were transfected with pCDNA-GFP vector or pCDNA-GFP-FAM47E construct and after 48 h of transfection, the cells were lysed in lysis buffer (10 mM Tris [pH 7.5], 150 mM NaCl, 0.5% NP40, 0.5 mM EDTA, and Protease inhibitor cocktail). The lysates were centrifuged at 18,400$g$ for 10 min at 4°C. The supernatant thus collected was labeled as soluble fraction and the pellet was resuspended in digestion buffer (10 mM Tris [pH 7.5], 150 mM NaCl, 0.5% NP40, 1.5 mM MgCl$_2$, protease inhibitor cocktail, and benzonase nuclease [Sigma-Aldrich]) and incubated in ice for 45 min. The benzonase digestion was stopped by adding 2 mM EDTA, reaction mixtures were centrifuged at 21,100$g$ for 20 min at 4°C and the supernatants were collected and labeled as chromatin fractions. The soluble fractions were immunoblotted and probed with anti-PRMT5 antibody or anti-$\beta$ actin antibody and the chromatin fractions were immunoblotted and probed with anti-PRMT5 antibody or anti-histone 3 antibody (Cat. no. ab1791; Abcam). The band intensities were quantified using the ImageJ software.

### Investigation of histone arginine methylation modifications

To investigate the effect of ectopic expression of FAM47E on histone arginine methylation modifications, HEK293 cells were transfected with pCDNA-GFP vector or pCDNA-GFP-FAM47E construct or control siRNA or FAM47E siRNA. After 48 h of transfection, the cells were harvested and histones were isolated from these cells by standard acid extraction method as detailed previously (Shechter et al, 2007). The isolated histones were resolved in 16% SDS–PAGE and immunoblotted and probed with anti-H4R3me2s antibody (Cat. no. ab5823; Abcam) or anti-H3R2me2s antibody (Cat. no. ab194684; Abcam) or anti-H3R8me2s (Cat. no. ab130740; Abcam) or anti-histone 3 antibody (Cat. no. ab1791; Abcam). The band intensities were quantified using the ImageJ software.

## qRT-PCR analysis of PRMT5 target genes expression and FAM47E isoforms

HEK293 cells were transfected with pCDNA-GFP vector or pCDNA-GFP-FAM47E construct or control siRNA or FAM47E siRNA or co-transfected with pCDNA-GFP-FAM47E construct and control siRNA or PRMT5 siRNA. After 48 h of post-transfection, the cells were harvested, total RNAs were extracted using Trizol reagent (Invitrogen) according to manufacturer's protocol and the cDNAs were prepared using Maxima H Minus Reverse Transcriptase (Thermo Fisher Scientific). The qRT-PCR analyses were performed using SYBR green Master mix (Roche) as per manufacturer's protocol. The qRT-PCR reactions were performed in triplicates and each assay was repeated at least three times. The expression of target genes was normalized to the expression of GAPDH. To quantify the expression of different FAM47E isoforms, the total RNA was isolated from HEK293 cells and qRT-PCR analyses were performed as described above using isoform specific primers. The primers used in the qRT-PCR are listed in Table S1.

## Chromatin immunoprecipitation

The chromatin was prepared from HEK293 cells as described previously (Awasthi et al, 2018). For the chromatin immunoprecipitation of PRMT5, 25 μg of chromatin was incubated with 4 μg of normal mouse IgG (Cat. no. sc-2025; Santa Cruz Biotechnology) or 4 μg of anti-PRMT5 antibody (Cat. no. sc-376937; Santa Cruz Biotechnology) for 2 h at 4°C. After the incubation, 25 μl of protein G beads were added to the tubes and incubated at 4°C with rotation for further 8 h. For the chromatin immunoprecipitation of FAM47E, 25 μg of chromatin was incubated with 4 μg of normal rabbit IgG (Cat. no. 2729; Cell Signaling Technology) or 4 μg of anti-FAM47E antibody (Cat. no. PA5-46681; Thermo Fisher Scientific) for 2 h at 4°C. After the incubation, 25 μl of protein A beads were added to the tubes and incubated at 4°C with rotation for further 8 h. Then the beads were washed extensively and DNA was purified as detailed previously (Awasthi et al, 2018). The qRT-PCR analyses of the immunoprecipitated DNA samples were carried out using SYBR green master mix (Roche) as described above. The primers used in the qRT-PCR are listed in Table S1.

## Methylation studies

HEK293 cells were transfected with pEGFP-SmD3 construct or pCDNA-GFP-FAM47E construct and treated with or without PRMT5 inhibitor, EPZ015666 (Sigma-Aldrich) to investigate the methylation of FAM47E by PRMT5. The pEGFP-SmD3 construct was co-transfected with pCDNA-HA vector or pCDNA-HA-FAM47E construct in HEK293 cells to study the influence of FAM47E on the methylation of SmD3 by PRMT5. After 48 h of transfection, the cells were lysed in the lysis buffer (10 mM Tris [pH:7.5] 150 mM NaCl, 0.5 mM EDTA, 0.5% NP-40, and protease inhibitor cocktail; Roche) and incubated with GFP-Trap A beads (ChromoTek) for 2 h at 4°C. After the incubation, the beads were washed extensively with lysis buffer and bound fractions were eluted by boiling the beads at 100°C for 5 min and the eluted proteins were separated in 12% SDS–PAGE, transferred to PVDF membrane and probed using or anti symmetric dimethyl arginine antibody, SYM10 (Cat. no. 07-412; Merck Millipore), or anti-GFP antibody (Cat. no. 632375; Clontech).

## Immunofluorescence studies

For immunofluorescence, HEK293 cells were grown on the coverslip up to 80% confluency. Then the cells were washed with PBS; fixed by using 4% formaldehyde and permeabilized with PBS containing 0.25% Triton X-100. Then the cells were incubated with anti-FAM47E antibody (Cat. no. PA5-46681; Thermo Fisher Scientific) for overnight at 4°C and probed with anti-rabbit IgG Dylight 633 antibody (Cat. no. 35563; Thermo Fisher Scientific). Then the cells were stained with DAPI and embedded using the Mowiol mounting medium, and the images were taken using the confocal microscope (LSM 510 Meta instrument).

## Cell proliferation and clonogenic assays

HeLa cells were transfected with pCDNA-GFP vector or pCDNA-GFP-FAM47E construct or control siRNA or FAM47E siRNA or co-transfected with pCDNA-GFP-FAM47E construct and control siRNA or PRMT5 siRNA. After 36, 48, 72 and 96 h of transfection, the cells were harvested, stained with trypan blue, and counted using a hemocytometer. MTT assays were performed as described previously (Awasthi et al, 2018).

HeLa cells were transfected with pCDNA-GFP vector or pCDNA-GFP-FAM47E construct or control siRNA or FAM47E siRNA or co-transfected with pCDNA-GFP-FAM47E construct and control siRNA or PRMT5 siRNA. After 24 h of transfection, the clonogenic assay was carried out as described earlier (Awasthi et al, 2018).

# Supplementary Information

# Acknowledgements

This work was funded by Innovative Young Biotechnologist Award, Department of Biotechnology, Government of India (Grant No BT/03/IYBA/2010; A Dhayalan; Ramalingaswami Re-entry Fellowship: BT/RLF/Re-entry/05/2018; S Chavali; Research Associateship; RV Kadumuri), Board of Research in Nuclear Sciences (Grant No 37(1)/14/17/2017-BRNS/37019; A Dhayalan), Science & Engineering Research Board (Grant No. SRG/2019/001785; S Chavali), Council of Scientific and Industrial Research (Junior and Senior Research Fellowships to B Chakrapani and S Awasthi), University Grants Commission (UGC), Government of India (Junior and Senior Basic Scientific Research Fellowships to MIK Khan and A Mahesh), Pondicherry University (PhD student fellowship to S Gupta and M Verma), Rajiv Gandhi Centre for Biotechnology Intramural grant (A Rajavelu), and IISER Tirupati (Post-doc fellowship to RV Kadumuri; Intramural support to S Chavali). We acknowledge Fund for Improvement of S&T Infrastructure in Universities and Higher Educational Institutions Program of Department of Science and Technology (DST-FIST) and UGC-Special Assistant Programme that funded instrumentation facilities of Department of Biotechnology, Pondicherry University.

## Author Contributions

B Chakrapani: conceptualization, formal analysis, validation, investigation, visualization, methodology, and writing—original draft.

MIK Khan: conceptualization, formal analysis, validation, investigation, methodology, and writing—original draft.

RV Kadumuri: data curation, formal analysis, investigation, visualization, and writing—original draft.

S Gupta: validation, investigation, and methodology.

M Verma: validation, investigation, and methodology.

S Awasthi: validation, investigation, and methodology.

G Govindaraju: validation and investigation.

A Mahesh: validation, investigation, and methodology.

A Rajavelu: conceptualization, investigation, and writing—original draft.

S Chavali: conceptualization, data curation, formal analysis, supervision, funding acquisition, validation, investigation, visualization, methodology, and writing—original draft, review, and editing.

A Dhayalan: conceptualization, resources, formal analysis, supervision, funding acquisition, validation, investigation, visualization, methodology, project administration, and writing—original draft, review, and editing.

## Conflict of Interest Statement

The authors declare that they have no conflict of interest.

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
