## [Reviewer comments · Life Science Alliance]

Life Science Alliance

The uncharacterized protein FAM47E interacts with PRMT5 and regulates its functions

Baskar Chakrapani, Mohd. Imran Khan, Rajashekar Kadumuri, Somlee Gupta, Mamta Verma, Sharad Awasthi, Gayathri Govindaraju, Arun Mahesh, Arumugam Rajavelu, Sreenivas Chavali, and Arunkumar Dhayalan

DOI: <https://doi.org/10.26508/lsa.202000699>

Corresponding author(s): Arunkumar Dhayalan, Pondicherry University, India and Sreenivas Chavali, Indian Institute of Science Education and Research, Tirupati, India

Review Timeline:

Submission Date:	2020-03-13
Editorial Decision:	2020-04-29
Revision Received:	2020-12-09
Editorial Decision:	2020-12-16
Revision Received:	2020-12-18
Accepted:	2020-12-18

Scientific Editor: Shachi Bhatt

Transaction Report:

April 29, 2020

Re: Life Science Alliance manuscript #LSA-2020-00699-T

Dr. Arunkumar Dhayalan
Pondicherry University, India
Department of Biotechnology
Department of Biotechnology
Pondicherry University
Puducherry, Puducherry 605014
India

Dear Dr. Dhayalan,

Thank you for submitting your manuscript entitled "The Uncharacterized protein, FAM47E interacts with PRMT5 and regulates its functions" to Life Science Alliance. The manuscript was assessed by expert reviewers, whose comments are appended to this letter.

As you will see, your work received somewhat split views from the reviewers. While reviewer #1 thinks that only a more minor revision is necessary, reviewer #2 thinks that the observed effects are small and that your conclusions need better support. We think it is overall rather straightforward to address both reviewers' criticisms and that doing so will indeed strengthen your conclusions and thus your manuscript. The main point of disagreement between the reviewers that crystallized during our reviewer cross-commenting session relates to the request to consolidate the knock-down-based findings with a different (knock-out) approach. We concluded that a re-analysis of the effects upon knock-down and a slightly more extensive analysis (points 3-7 of rev#2) to substantiate your conclusions is sufficient. We would thus like to invite you to submit a revised version of your manuscript, addressing the requests of rev#1 as well as those mentioned above to address the concerns of rev#2. In our view these revisions should typically be achievable in around 3 months. However, we are aware that many laboratories cannot function fully during the current COVID-19/SARS-CoV-2 pandemic and therefore encourage you to take the time necessary to revise the manuscript to the extent requested above. We will extend our 'scoping protection policy' to the full revision period required. If you do see another paper with related content published elsewhere, nonetheless contact me immediately so that we can discuss the best way to proceed.

Please note that papers are generally considered through only one revision cycle, so strong support from the referees on the revised version is needed for acceptance.

Thank you for this interesting contribution to Life Science Alliance. We are looking forward to receiving your revised manuscript.

Sincerely,

B. MANUSCRIPT ORGANIZATION AND FORMATTING:

Reviewer #1 (Comments to the Authors (Required)):

The authors have screened for interactors of PRMT5 and identified FAM47E in a Y2H screen. With pulldown experiments they demonstrated the direct interaction of both proteins under cellular conditions. Next they showed that binding of FAM47E increases the protein stability and chromatin association of PRMT5, the abundance of PRMT5 introduced histone methylation and corresponding gene regulation. Finally they show that FAM47E increases the proliferation potential of cells. This is a very interesting paper. Experiments have been conducted in a very solid manner with all necessary controls included. Data are very convincing and potentially very important.

Comments:

1) I have one content related question, which is that readers would like to know if FAM47E is methylated by PRMT5.

Moreover, I have some minor suggestions regarding the text and presentation

2) Remove comma in the title.

3) In the abstract add comma after FAM47E "uncharacterized protein, FAM47E, as an..."

4) p. 4: "has higher enzymatic activity than PRMT5 in the unbound state"

5) p. 5 and several places: In my view it should read "low stringency" and "high stringency" media.

6) I do not think it necessary to labelled empty lanes in the gel images.

7) Fig. 1A: Explain somewhere the setup of the assay. What are the media supplements listed in 1A good for?

8) Fig. 1: It would be good to indicate in all pulldowns what was pulled. Currently this is sometimes mentioned in the heading sometimes not. Change the "IP" label by a more specific one like "GFP trap", "GST pulldown",...

9) Many figures: The GFP bands in the control GFP expression should be visible in anti-GFP blots. This can be shown in the supplement at least for some selected examples.

10) Fig. 4C is difficult to read. Can the authors try to sort be condition and show one plot with all samples and controls for each condition?

Reviewer #2 (Comments to the Authors (Required)):

Summary:

In this study, Baskar Chakrapani et. al. convincingly demonstrate that PRMT5 interacts with a as yet uncharacterized protein called FAM47E. The PRMT5/FAM47E interaction was discovered using a Y2H approach. They go on to show that the stability of PRMT5 is likely regulated by FAM47E. Also, FAM47E seems to play a role in recruiting PRMT5 to chromatin and enhancing a number of arginine methylation marks on histones.

Critique:

This manuscript is well-written and easy to read. However, the findings are rather preliminary. A general concern with this study is that the knock-down efficiency of PRMT5 and FAM47E is not very good (generally around 50%), and the authors should consider doing CRISPR-mediated KO of FAM47E to solidify their story. There are a number of additional points to address, which I highlight below.

Issues:

- 1) An important issue to resolve is if FAM47E also interacts with MEP50. MEP50 is a very stable component of the PRMT5 complex. Does FAM47E evict and replace MEP50?
- 2) The effects of FAM47E knockdown and overexpression on PRMT5 stability are very subtle. If a FAM47E knockout was used, the impact on PRMT5 stability may be more obvious.
- 3) They have performed fractionation experiments looking at FAM47E in the soluble nuclear fraction vs the chromatin associated nuclear fraction. Is FAM47E only found in the nucleus, or is it also a cytoplasmic protein? In other words, does it only regulate PRMT5's chromatin functions?
- 4) Related to this issue, the investigators need to look at additional PRMT5 substrates. There are good SDMA antibodies to SmB/B', and good semi-pan SDMA antibodies, that will facilitate these experiments.
- 5) In Figure 4C, they show that FAM47E KD or OE can impact the expression of a few PRMT5 regulated genes. Does FAM47E ChIP at these promoters, along with PRMT5.
- 6) In the discussion they mention "...FAM47E is the first binding partner that regulates PRMT5 stability ...". This may not be true, as MEP50 is critical for PRMT5 stability. If MEP50 is knocked out, there is almost total loss of PRMT5. However, I am not sure if this PRMT5 reduction is due to proteasomal degradation. This issue should be addressed in the discussion.
- 7) Finally, they need to closely look at the Huan-Tian Zhang paper (BBA, 2016). I think this study was the first to show that PRMT5 is a target for proteasomal degradation. They superficially reference this manuscript in the introduction, but they do not link any of their studies to this published report. Specifically, the BBA paper shows that CHIP is an E3 for PRMT5. Does the FAM47E interaction with PRMT5 block the ability of CHIP to interact with PRMT5, thereby stabilizing it?

Response to Reviewer's Comments

Reviewer #1

Comment 1.0: The authors have screened for interactors of PRMT5 and identified FAM47E in a Y2H screen. With pulldown experiments they demonstrated the direct interaction of both proteins under cellular conditions. Next they showed that binding of FAM47E increases the protein stability and chromatin association of PRMT5, the abundance of PRMT5 introduced histone methylation and corresponding gene regulation. Finally they show that FAM47E increases the proliferation potential of cells. This is a very interesting paper. Experiments have been conducted in a very solid manner with all necessary controls included. Data are very convincing and potentially very important.

Response 1.0: We thank the reviewer for the positive evaluation of our manuscript and for the insightful comments, addressing which has strengthened our manuscript further.

Comments 1.1: I have one content related question, which is that readers would like to know if FAM47E is methylated by PRMT5.

Response 1.1: As suggested by the reviewer, we investigated if FAM47E is a methylation substrate of PRMT5. For this, we overexpressed GFP-FAM47E and a known PRMT5 substrate, GFP-SmD3, as positive control, in HEK293 cells in the presence and absence of PRMT5 specific inhibitor, EPZ015666 (Chan-Penebre et al, 2015). GFP-FAM47E and GFP-SmD3 proteins were immunoprecipitated from these cells and their methylation status was investigated using pan symmetric dimethyl arginine antibody. We observed a strong methylation signal in SmD3 protein and this signal was reduced in the SmD3 protein, which was isolated from the cells which were treated with EPZ015666. However, we could not detect any methylation signal with FAM47E protein, both in the presence and absence of EPZ015666, suggesting that FAM47E is unlikely to be a substrate of PRMT5 in the conditions that we had tested.

We have added this data in the Supplementary Figure S3 of the revised manuscript and have discussed these results in 'FAM47E enhances the stability of PRMT5 protein' sub-section of the Results section of the revised manuscript (Page 8, paragraph 1, line 1).

Comment 1.2: Moreover, I have some minor suggestions regarding the text and presentation

Remove comma in the title.

Response 1.2: We have removed the comma in the title of the revised version of the manuscript.

Comment 1.3: In the abstract add comma after FAM47E "uncharacterized protein, FAM47E, as an..."

Response 1.3: We have now added the comma in that sentence and the modified text now reads as “Here, using yeast two-hybrid screening, followed by immunoprecipitation and pull-down assays, we identify a previously uncharacterized protein, FAM47E, as an interaction partner of PRMT5.” in the revised manuscript (Page 3, paragraph 1, line 7).

Comment 1.4: p. 4: "has higher enzymatic activity than PRMT5 in the unbound state"

Response 1.4: We thank the reviewer for helping us increase the clarity in the statement. We have now modified the sentence which now reads as “For instance, PRMT5 forms a hetero-octameric complex with WD40 repeat protein, MEP50 and the PRMT5-MEP50 complex has higher enzymatic activity than PRMT5 in the unbound state.” in the revised manuscript (Page 5, paragraph 1, line 4).

Comment 1.5: p. 5 and several places: In my view it should read "low stringency" and "high stringency" media.

Response 1.5: We thank the reviewer for these corrections. We have now modified the terms “low stringent” and “high stringent” into “low stringency” and “high stringency” respectively in the revised manuscript (Page 6, paragraph 1, lines 3, 4 & 20).

Comment 1.6: I do not think it necessary to labelled empty lanes in the gel images.

Response 1.6: As suggested by the reviewer, we have removed the labels of the empty lanes in all the gel images. This has helped the gel images look clear and conveys the information better.

Comment 1.7: Fig. 1A: Explain somewhere the setup of the assay. What are the media supplements listed in 1A good for?

Response 1.7: We thank the reviewer of pointing this out. We have now detailed this information in the ‘Y2H Screening’ sub-section of the ‘Materials and Methods’ section of the revised manuscript (Page 16, paragraph 1, line 1). In addition, we have also described the media supplements in the legend of Fig. 1A in the revised manuscript to make it easy for the reader to follow the experiment (Page 32, paragraph 1, line 5).

Comment 1.8: Fig. 1: It would be good to indicate in all pulldowns what was pulled. Currently this is sometimes mentioned in the heading sometimes not. Change the "IP" label by a more specific one like "GFP trap", "GST pulldown".

Response 1.8: We thank the reviewer for helping us make this explicit in the figures. We have now mentioned clearly the specific pull down that was done as a title for each one of the respective panels in the Figure 1 in the revised manuscript. We have also replaced the IP label with the more specific terms in the figures.

Comment 1.9: Many figures: The GFP bands in the control GFP expression should be visible in anti-GFP blots. This can be shown in the supplement at least for some selected examples.

Response 1.9: We have now replaced the cropped anti-GFP blots in Figure 2A, Figure 2C and Figure 3A with uncropped blots to show the both GFP and GFP-FAM47E bands.

Comment 1.10: Fig. 4C is difficult to read. Can the authors try to sort by condition and show one plot with all samples and controls for each condition?

Response 1.10: We thank the reviewer for highlighting the difficulty in reading this figure. As per the reviewer’s suggestion, we have now split the plots based on the condition and have provided the data in the revised Figure 5B (Initial submission Fig. 4C). We believe that this re-organization helps read the figure better and facilitates easy interpretation of the conclusions drawn from the data presented in the figure.

Reviewer #2:

Comment 2.0: Summary: In this study, Baskar Chakrapani et. al. convincingly demonstrate that PRMT5 interacts with a as yet uncharacterized protein called FAM47E. The PRMT5/FAM47E interaction was discovered using a Y2H approach. They go on to show that the stability of PRMT5 is likely regulated by FAM47E. Also, FAM47E seems to play a role in recruiting PRMT5 to chromatin and enhancing a number of arginine methylation marks on histones.

Critique: This manuscript is well-written and easy to read. However, the findings are rather preliminary. A general concern with this study is that the knock-down efficiency of PRMT5 and FAM47E is not very good (generally around 50%), and the authors should consider doing CRISPR-mediated KOs of FAM47E to solidify their story. There are a number of additional points to address, which I highlight below.

Response 2.0: We thank the reviewer for the positive evaluation of our manuscript and for providing critical inputs that has helped to expand the scope of our work and consolidate our findings.

We agree with the reviewer that the CRISPR-mediated knock out of FAM47E would have been an ideal option. But, we have completed the entire work by perturbing FAM47E levels through siRNA mediated knockdown and over-expression. Though the knock-down efficiency is generally around 50%, we could still observe the effects of FAM47E knockdown on the levels of PRMT5 and other PRMT5-mediated functions. We agree that if the knockdown efficiency is more, then the observed effects might have been quantitatively higher. Nevertheless, the partial loss is substantial enough to capture the impact of FAM47E interaction with PRMT5. Therefore, we reckon that efficiency of knock down will not alter our conclusions and interpretations. Since the reported conclusions and interpretations are proved beyond any reasonable doubts with siRNA-mediated knockdown of FAM47E and the overexpression of FAM47E, we hope that the reviewer would find merit in our data and the corresponding interpretations.

Comment 2.1: An important issue to resolve is if FAM47E also interacts with MEP50. MEP50 is a very stable component of the PRMT5 complex. Does FAM47E evict and replace MEP50?

Response 2.1: We thank the reviewer for raising these points. To address these, we first investigated whether FAM47E also interacts with MEP50 in addition to PRMT5. For this, we performed co-immunoprecipitation by co-expressing GFP or GFP-FAM47E with Myc-tagged MEP50 in HEK293 cells and found that FAM47E interacts with MEP50. We also investigated if FAM47E affects the binding of MEP50 with PRMT5. For this, we performed co-immunoprecipitation by co-expressing GFP or GFP-PRMT5 and Myc-tagged MEP50 with or without HA tagged FAM47E in HEK293 cells. We observed that the over-expression of FAM47E did not affect the PRMT5-MEP50 interaction suggesting that FAM47E interacts with PRMT5 and MEP50 without affecting the PRMT5-MEP50 complex.

We have added these results in Supplementary Figure S2 of the revised manuscript and have discussed these results in ‘PRMT5 interacts with FAM47E’ sub-section of the Results section of the revised manuscript (Page 7, paragraph 2, line 1).

Comment 2.2: The effects of FAM47E knockdown and overexpression on PRMT5 stability are very subtle. If a FAM47E knockout was used, the impact on PRMT5 stability may be more obvious.

Response 2.2: As we mentioned in our response 2.0, knock-out would have been an ideal option. However, at this point in time, it is not possible for us to repeat the whole study by generating and using FAM47E knockout. Moreover, we observed an increase of ~2.3 folds in the protein levels of PRMT5 upon the over-expression of FAM47E (Figure 2A). The knock-down of FAM47E reduced the PRMT5 protein levels by ~ 39% (Figure 2B). These effects are substantial enough to justify the interpretations and conclusions that FAM47E enhances the stability of PRMT5 protein.

Comment 2.3: They have performed fractionation experiments looking at FAM47E in the soluble nuclear fraction vs the chromatin associated nuclear fraction. Is FAM47E only found in the nucleus, or is it also a cytoplasmic proteins? In other words, does it only regulate PRMT5's chromatin functions?

Response 2.3: To test the sub-cellular localization of FAM47E, we performed immunofluorescence studies. We found that FAM47E is distributed both in the cytoplasm and nucleus (Supplementary Figure S9). Since FAM47E and PRMT5 are present in both cytoplasm and nucleus, FAM47E might regulate the non-chromatin functions of PRMT5 as well (Page 11, paragraph 2, line 11). Please refer Response 2.4 for our experiments investigating the role of FAM47E on the regulation of non-chromatin functions of PRMT5.

Comment 2.4: Related to this issue, the investigators need to look at additional PRMT5 substrates. There are good SDMA antibodies to SmB/B', and good semi-pan SDMA antibodies, that will facilitate these experiments.

Response 2.4: To investigate whether FAM47E affects the non-chromatin functions of PRMT5, we analyzed the methylation level of SmD3, a well-studied non-chromatin substrate of PRMT5 (Meister et al, 2001; Friesen et al, 2001) upon over-expression of FAM47E. For this, we over-expressed GFP-SmD3 with or without FAM47E-HA in HEK293 cells. GFP-SmD3 was immunoprecipitated from these cells and its methylation levels were analyzed by using pan symmetric dimethyl arginine antibody, SYM10. The reliability of SYM10 antibody in detecting PRMT5 mediated methylation of SmD3 was confirmed by analyzing the methylation levels of SmD3 in the cells which were treated with or without PRMT5 inhibitor, EPZ015666 (Chan-Penebre et al, 2015) (Supplementary Figure S3). We found that the over-expression of FAM47E did not alter methylation levels of SmD3 suggesting that FAM47E primarily affects the epigenetic functions of PRMT5 (Supplementary Figure S10). Nevertheless, this does not rule out the plausible role of FAM47E in affecting other non-chromatin functions of PRMT5. Further detailed investigations are required to unravel this aspect.

We have added these results in Supplementary Figure S10 of the revised manuscript and have discussed these results in 'FAM47E promotes the chromatin association of PRMT5 and histone arginine methylation' sub-section of the Results section of the revised manuscript (Page 11, paragraph 2, line 11).

Comment 2.5: In Figure 4C, they show that FAM47E KD or OE can impact the expression of a few PRMT5 regulated genes. Does FAM47E ChIP at these promoters, along with PRMT5.

Response 2.5: We thank the reviewer for this insightful input. As suggested by the reviewer, we investigated the binding of FAM47E and PRMT5 at the promoters of the tested PRMT5 target genes by using chromatin immunoprecipitation. We found that both FAM47E and PRMT5 proteins were enriched at the promoter regions of the tested PRMT5 target genes indicating that the FAM47E binds to the promoters of the PRMT5 target genes along with PRMT5 and contributes for the PRMT5-mediated epigenetic regulation.

We have added these data in Fig. 5C of the revised manuscript and have discussed these results in ‘FAM47E promotes the chromatin association of PRMT5 and histone arginine methylation’ sub-section of the Results section of the revised manuscript (Page 11, paragraph 2, line 1).

Comment 2.6: In the discussion they mention "...FAM47E is the first binding partner that regulates PRMT5 stability ...". This may not be true, as MEP50 is critical for PRMT5 stability. If MEP50 is knocked out, there is almost total loss of PRMT5. However, I am not sure if this PRMT5 reduction is due to proteasomal degradation. This issue should be addressed in the discussion.

Response 2.6: We thank the reviewer for pointing this out. We have now included the details about the PRMT5-MEP50 interaction mediated regulation of PRMT5 levels and have modified the text in revised manuscript.

The modified text now reads as “On the other hand, several interaction partners of PRMT5 and their functional outcomes have been extensively studied. Specifically, MEP50 forms an octameric complex with PRMT5 and regulates its enzymatic activity and its levels (Friesen et al, 2002; Antonyamy et al, 2012; Ho et al, 2013; Saha & Eckert, 2015; Gonsalvez et al, 2007; Saha et al, 2016; Chen et al, 2017). Strikingly, here we report that FAM47E increases the stability of PRMT5 and enhances its chromatin methylation activity (Figure 7). While, both the proteins seem to interact with PRMT5 and have overlapping effects, these could represent two distinct modes of regulation and function for PRMT5. For instance, here we show that FAM47E affects the stability of PRMT5 by inhibiting its proteasomal degradation. However, the mechanisms by

which MEP50 regulates the levels of PRMT5 is unknown. In terms of functional impact, FAM47E enhances the chromatin methylation activity of PRMT5, by increasing its association with chromatin. On the other hand, MEP50 enhances the enzymatic activity of PRMT5 by increasing its affinity towards the substrate and the cofactor (Antonyamy et al, 2012). Further research is required to delineate the molecular mechanisms underlying the regulation and functional outcomes of PRMT5 upon binding with FAM47E and MEP50.” (Page 13, paragraph 2, line 13).

Comment 2.7: Finally, they need to closely look at the Huan-Tian Zhang paper (BBA, 2016). I think this study was the first to show that PRMT5 is a target for proteasomal degradation. They superficially reference this manuscript in the introduction, but they do not link any of their studies to this published report. Specifically, the BBA paper shows that CHIP is an E3 for PRMT5. Does the FAM47E interaction with PRMT5 block the ability of CHIP to interact with PRMT5, thereby stabilizing it?

Response 2.7: We thank the reviewer for highlighting this point. We have now discussed our findings in the context of the previous report (Zhang et al., 2016) in the revised manuscript (Page 8). As suggested by the reviewer, we investigated whether FAM47E-PRMT5 interaction affects the binding of the E3 ubiquitin ligase, CHIP with PRMT5. For this, we co-expressed GFP or GFP-PRMT5 and Myc-tagged CHIP with or without HA tagged FAM47E. We observed that the over-expression of FAM47E did not affect the PRMT5-CHIP interaction but on the contrary the over-expression of FAM47E enhanced PRMT5-CHIP interaction mildly (Supplementary Fig. S7). The mild enhancement of PRMT5-CHIP interaction might be due to the increase in the protein levels of PRMT5 upon the over-expression of FAM47E.

This suggest that the stabilization of PRMT5 by FAM47E is not mediated by disrupting the PRMT5-CHIP interaction. However, this does not rule out the possibility that FAM47E-PRMT5 interaction might block the CHIP mediated polyubiquitination of PRMT5. Since the PRMT5 is ubiquitinated at multiple lysine residues (Zhang et al, 2016), it is also possible that FAM47E-PRMT5 interaction might inhibit the polyubiquitination of PRMT5 mediated by as yet unidentified E3 ubiquitin ligase(s) that targets PRMT5. Our results lay foundation for future

investigations to delineate the mechanisms underlying FAM47E inhibition of the proteasomal degradation of PRMT5.

We have added these data in Supplementary Figure S7 of the revised manuscript and have discussed these results in ‘FAM47E enhances the stability of PRMT5 protein’ sub-section of the Results section of the revised manuscript (Page 9, paragraph 2, line 1).

December 16, 2020

RE: Life Science Alliance Manuscript #LSA-2020-00699-TR

Dr. Arunkumar Dhayalan
Pondicherry University, India
Department of Biotechnology
Department of Biotechnology
Pondicherry University
Puducherry, Puducherry 605014
India

Dear Dr. Dhayalan,

Thank you for submitting your revised manuscript entitled "The uncharacterized protein FAM47E interacts with PRMT5 and regulates its functions". We would be happy to publish your paper in Life Science Alliance pending final revisions necessary to meet our formatting guidelines.

Along with the points listed below, please also attend to the following:

- please add ORCID ID for secondary corresponding author-they should have received instructions on how to do so
- please provide source data (uncropped, unedited gel images) for blots shown in Figure 1F and S3
- please re-label Figure 4 to organize it in panels, to improve readability. please also update the figure legend and callouts in the manuscript accordingly
- please add scale bars to Figure 6D

A. FINAL FILES:

B. MANUSCRIPT ORGANIZATION AND FORMATTING:

Sincerely,

Shachi Bhatt, Ph.D.
Executive Editor
Life Science Alliance
<https://www.lsjournal.org/>
Tweet @SciBhatt @LSAJournal

Reviewer #1 (Comments to the Authors (Required)):

All my comments have been addressed in a very convincing manner.

December 18, 2020

RE: Life Science Alliance Manuscript #LSA-2020-00699-TRR

Dr. Arunkumar Dhayalan
Pondicherry University, India
Department of Biotechnology
Department of Biotechnology
Pondicherry University
Puducherry, Puducherry 605014
India

Dear Dr. Dhayalan,

Thank you for submitting your Research Article entitled "The uncharacterized protein FAM47E interacts with PRMT5 and regulates its functions". It is a pleasure to let you know that your manuscript is now accepted for publication in Life Science Alliance. Congratulations on this interesting work.

*****IMPORTANT:** If you will be unreachable at any time, please provide us with the email address of an alternate author. Failure to respond to routine queries may lead to unavoidable delays in publication.*******

DISTRIBUTION OF MATERIALS:

Again, congratulations on a very nice paper. I hope you found the review process to be constructive and are pleased with how the manuscript was handled editorially. We look forward to future exciting

submissions from your lab.

Sincerely,

Shachi Bhatt, Ph.D.

Executive Editor

Life Science Alliance

<https://www.lsjournal.org/>
